# Novel flight style and light wings boost flight performance of tiny beetles

Sergey E. Farisenkov[1,7 ✉], Dmitry Kolomenskiy[2,3,7], Pyotr N. Petrov[1], Thomas Engels[4], Nadezhda A. Lapina[1], Fritz-Olaf Lehmann[4], Ryo Onishi[2], Hao Liu[5] & Alexey A. Polilov[1,6 ✉]

Flight speed is positively correlated with body size in animals[1]. However, miniature featherwing beetles can fly at speeds and accelerations of insects three times their size[2]. Here we show that this performance results from a reduced wing mass and a previously unknown type of wing-motion cycle. Our experiment combines three-dimensional reconstructions of morphology and kinematics in one of the smallest insects, the beetle *Paratuposa placentis* (body length 395 µm). The flapping bristled wings follow a pronounced figure-of-eight loop that consists of subperpendicular up and down strokes followed by claps at stroke reversals above and below the body. The elytra act as inertial brakes that prevent excessive body oscillation. Computational analyses suggest functional decomposition of the wingbeat cycle into two power half strokes, which produce a large upward force, and two down-dragging recovery half strokes. In contrast to heavier membranous wings, the motion of bristled wings of the same size requires little inertial power. Muscle mechanical power requirements thus remain positive throughout the wingbeat cycle, making elastic energy storage obsolete. These adaptations help to explain how extremely small insects have preserved good aerial performance during miniaturization, one of the factors of their evolutionary success.

Driven by curiosity about the smallest objects, scientific exploration of the microscopic world has facilitated the miniaturization of various industrial products. But miniaturization is not just a human-made artifice: success stories of miniaturization are abundant in the living world. For more than 300 million years, ecological pressures have forced insects to develop extremely small bodies down to 200 µm long[3] without losing their ability to fly. As the physical properties of flight depend on size, constraints that are insignificant at the macro scale become significant at the micro scale, and vice versa[4]. Compared with larger sizes, flight at small sizes is dominated by viscous air friction rather than inertial forces resulting from the acceleration of the surrounding air. This competition between friction and inertia is key for flight at all size scales and thus applies to all animals that move through air.

Large insects generally fly faster than smaller ones[1]. Nevertheless, some of the smallest insects fly surprisingly well. For example, it was recently revealed that minute featherwing beetles (Coleoptera: Staphylinoidea: Ptiliidae) typically fly with similar speeds to their larger relatives (Staphylinidae), despite a threefold difference in body length[2]. Moreover, ptiliids can accelerate twice as fast as carrion beetles (Staphylinoidea: Silphidae), although the latter are an order of magnitude larger. As the size-specific flight-muscle volume is smaller in Ptiliidae than in larger beetles[5], their excellent flight performance must result from the peculiar structure of their wings and flight style. Ptiliids have feather-like bristled wings—a condition known

as ptiloptery (Fig. 1b)—instead of the membranous wings possessed by most insects. This visually striking modification of the flight apparatus evolved convergently in extremely small representatives of several insect orders. The functional benefits of ptiloptery, however, have remained largely unknown.

Although many studies have focused on the secrets of flight in minute insects[6,7], most experimental data that elucidate wing motion and aerodynamics have been obtained from larger insect species[8–11]. Thus, unsteady aerodynamics of millimetre-size insects such as fruit flies[12,13] and mosquitoes[14] have received considerable attention in recent decades, whereas studies focusing on tiny insects remained scarce. Two-dimensional numerical studies on the aerodynamics of insect wings have previously shown that the flow past evenly spaced cylinder lattices reduces aerodynamic force production in bristled wings[15,16]. By contrast, experiments with mechanical comb-like models have suggested slightly larger lift-to-drag ratios during the clap-and-fling phase in bristled wings compared with membranous wings[17–19], but did not cover the full wingbeat cycle. Meanwhile, using state-of-the-art high-speed videography, it has become clear that small insects use a wingbeat cycle that is different from that of the larger ones[10,11], but, to our knowledge, the role of ptiloptery in this cycle has not been considered.

In this study, we analysed the flight of the miniature featherwing beetle *Paratuposa placentis*. We constructed a morphological model based

[1]Department of Entomology, Faculty of Biology, Lomonosov Moscow State University, Moscow, Russia. [2]Global Scientific Information and Computing Center, Tokyo Institute of Technology, Tokyo, Japan. [3]Skoltech Center for Design, Manufacturing and Materials, Skolkovo Institute of Science and Technology, Moscow, Russia. [4]Department of Animal Physiology, Institute of Biological Sciences, University of Rostock, Rostock, Germany. [5]Graduate School of Engineering, Chiba University, Chiba, Japan. [6]Joint Russian-Vietnamese Tropical Research and Technological Center, Southern Branch, Ho Chi Minh City, Vietnam. [7]These authors contributed equally: Sergey E. Farisenkov, Dmitry Kolomenskiy. ✉e-mail: farisenkov@entomology.bio.msu.ru; polilov@gmail.com

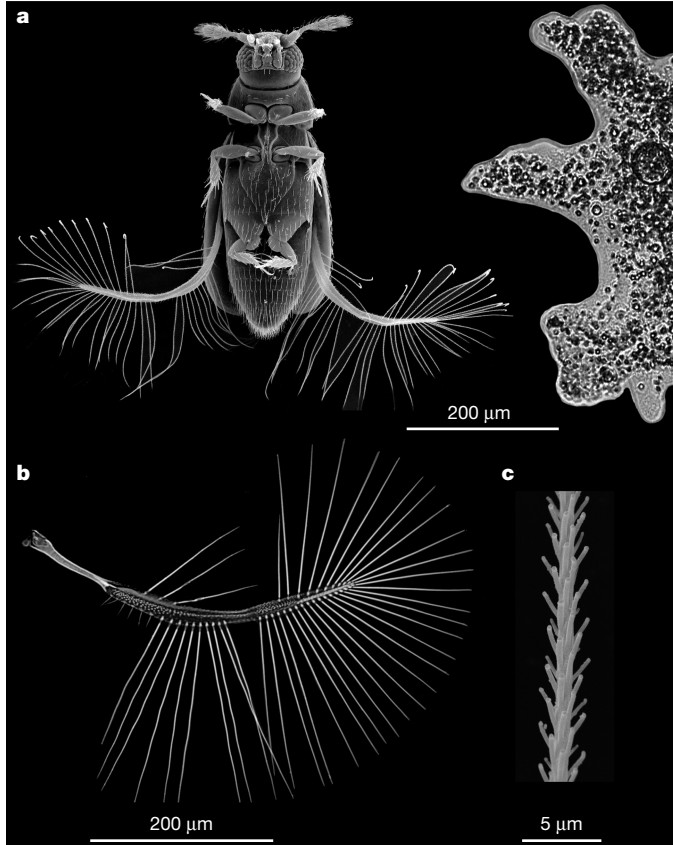

**Fig. 1 | External morphology of *P. placentis*. a–c**, Scanning electron microscopy images showing relative size of *P. placentis* (left) and *A. proteus* (right) (**a**), wing of *P. placentis* (**b**) and part of a seta (**c**). Every image was obtained from one randomly selected specimen; for detailed morphometry, see Supplementary Information.

on data gained from light, confocal and electron microscopy measurements, a kinematical model using synchronized high-speed videography, and a dynamical model using computational methods of solid and fluid mechanics. The combination of these methods offers a comprehensive view of how bristled wings work and explains why common sub-millimetre flying insects have bristled rather than membranous wings.

## Structural features of *P. placentis*

*P. placentis* is one of the smallest non-parasitic insect species, with a body length of about 395 ± 21 μm (all measurements are given as mean ± s.d.). This size is similar to the size of some unicellular protists such as *Amoeba proteus* (Fig. 1a). The body mass of *P. placentis* is 2.43 ± 0.19 μg (Supplementary Information). The bristled wing consists of a petiole, a narrow wing blade and a fringe of setae (bristles) covered with secondary outgrowths (Fig. 1b, c). The wing length is 493 ± 18 μm and the setae occupy 95.1 ± 0.3% of the aerodynamically effective wing area (interior of the green contour in Fig. 2b).

## Wing kinematics

The wingbeat cycle of *P. placentis* consists of two power strokes, during which most of the total flight force is generated[20], and two recovery strokes with wings clapping above and below the body (Fig. 2a, c, Supplementary Videos 1–6). Dorsal and ventral recovery strokes are unique to the Ptiliidae and replace the conventional clap-and-fling kinematics described in other insects, including miniature thrips[9] and parasitoid wasps[8]. Despite the large stroke amplitude, the wings do not

always clap tightly at the end of the ventral recovery stroke, depending on flight conditions (Supplementary Information). The setal fringes of the left and right wings may intersect during the fling phases of the recovery strokes. The morphological downstroke and upstroke are remarkably similar: the angle of attack (AoA) reaches 73° during the downstroke and 85° in magnitude during the upstroke (Fig. 2f). The cycle-averaged Reynolds number (Re) based on the mean speed of the radius of gyration is 9 and reaches 20 during power strokes when wing velocity is highest. The increased AoA during power strokes and the presence of recovery strokes are similar to the kinematics of swimming in miniature aquatic crustaceans (Supplementary Information), which move at similar flow regimes–for example, larvae of *Artemia* sp.[20], with a Reynolds number of 10.

## Vertical force generation

The wide rounded self-intersecting paths of the wing tips and dynamically changing orientation of the wings (Figs. 2c, 3a) maximize the aerodynamic asymmetry between power and recovery strokes. Upon each power stroke, geometrical AoA and wing velocity simultaneously reach their maxima (Fig. 2f). While forces and velocities are anti-aligned (Fig. 3a), their peaks are synchronized (Figs. 3d, 2f). The wing thus first produces an increased upward force as it quickly moves flat-on with net downward displacement and, subsequently, a small downward force while slowly moving edge-on upwards. The near-clap motion reduces the parasite downward force upon recovery[21]. Decomposition of the vertical force exerted on the wing into drag and lift (Methods) is shown in Fig. 3b, d. The vertical force due to drag exhibits greater positive peaks than that due to lift. This is accompanied by extended times of slightly negative drag-based vertical force. In association with these peaks, airflow simulation reveals a pair of strong vortex rings that are typical for drag-producing bodies (Fig. 3c, Supplementary Videos 5, 6). Approximately 32% of the cycle-averaged vertical force results from drag and 68% results from lift, indicating that the beetle benefits from both components. On average, the aerodynamic mechanisms produce bodyweight-supporting lift of 2.7 μg (Fig. 3g) (the beetle's estimated body mass is 2.4 μg) and a vertical acceleration of 1.0 m s⁻². The net contribution of body and elytra to the vertical force is negligible (Fig. 3g).

## Stabilizing role of the elytra

The unusually large horizontal and vertical excursion of the wings during flapping poses a peculiar flight dynamics problem. The forces are small during the recovery strokes but the moment arm relative to the centre of mass is large. This results in a pitching moment large enough to overturn the body around its pitching axis (Extended Data Fig. 7, Supplementary Information). To compensate for these moments in synchrony with wing flapping, the insect opens and closes the elytra with large amplitudes ($\psi_{max} - \psi_{min} = 52°$) compared to other flying beetles[22–24]. Figures 2e, 3f show that the elytra act as an inertial brake. At times between *t/T* of 0 and 0.3, the wings are raised in a dorsal position and produce nose-up torque. As soon as the wings start their downstroke, the elytra close, causing a nose-down recoil torque on the body. During ventral clapping, the wings produce nose-down torque and the elytra decelerate and reopen. We found that the elytra movements decrease the amplitude of body-pitching oscillation by approximately 50% compared with flight without elytra (Supplementary Information). It is thus likely that the inertial brake observed in *P. placentis* is a feature of ptiliid beetles flying at high wingbeat amplitudes and unique to their peculiar flying style.

## Bristled versus membranous wings

Numerical modelling suggests a wing mass in *P. placentis* of approximately 0.024 μg, which is about 1% of the body mass. By contrast,

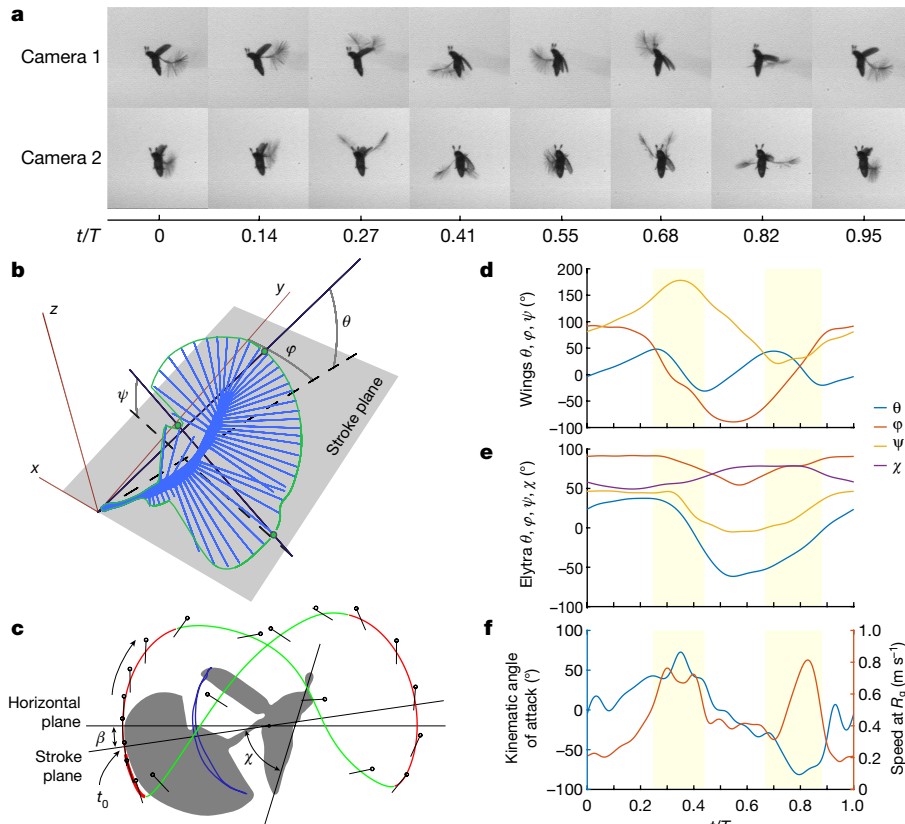

**Fig. 2 | Kinematics of *P. placentis*. a**, Frame sequence of a single stroke in two projections. **b**, Measurement scheme for Euler angles. **c**, Trajectory of the wing tip: recovery strokes (magenta line) and power strokes (green line) and measurement scheme for angle of body pitch (χ) and pitch of stroke plane to the horizon (β). **d**, Wing Euler angles as functions of dimensionless time $t/T$, $T = 1/f$: stroke deviation (θ), positional (φ) and pitch (ψ). **e**, Elytron Euler angles (θ, φ and ψ); body pitch angle (χ). **f**, AoA and wing speed at radius of gyration ($R_g$) versus $t/T$.

estimates of the mass of a membranous wing with the same outline amount to 0.13, 0.14 or 0.19 µg, depending on wing thickness (Supplementary Information). These estimates were based on some of the smallest membranous-winged insects, namely the wasp *Trichogramma telengai* (0.73 µm), the beetle *Orthoperus atomus* (0.85 µm), and the beetle *Limnebius atomus* (1.12 µm). *L. atomus* is closely related to the Ptiliidae. The maximum entry in the inertia matrix ($I_{zz}$) of the bristled wing is 1,600 µg µm², and for the membranous wings it is 13,800, 16,000 and 20,800 µg µm², respectively. Secondary outgrowths of the bristles are unique to Ptiliidae wings and reduce the wing mass by 44% compared with the bristled wing model with smooth cylindrical bristles at the same drag[25]. The bristled wing architecture with secondary outgrowths thus considerably reduces wing mass compared to a membranous architecture, while maintaining the needed aerodynamic properties. This conclusion is also supported by an allometric analysis of wing mass in differently sized insects (Extended Data Fig. 1, Supplementary Information).

Whereas instantaneous vertical forces generated by the simulated membranous wing outscore the bristled wing of *P. placentis* (Fig. 3d), the latter produces as much as 68% of the mean vertical force of the membranous wing. The vertical force peaks during power strokes produce similar-sized peaks in the mechanical power required for wing actuation. Cycle-averaged power consumption in *P. placentis* is relatively low and amounts to only 28 W per kg body mass, but instantaneous power may reach up to 110 W kg⁻¹ at $t/T = 0.82$ (Fig. 3e, h) owing to aerodynamic power. The total mechanical power of the bristled wing model (Fig. 3e) remains positive during the entire wing-beat cycle, because low inertia of the wing and high viscous damping of the surrounding air enable continuous energy transfer from the

flight apparatus to the wake. No elastic energy storage is required. By contrast, for a membranous wing, the inertial power is similar in peak magnitude to the aerodynamic power. Such a wing requires perfect elastic energy storage to achieve its minimum mean mechanical power of 37 W kg⁻¹ and powerful flight muscles to satisfy 180–210 W kg⁻¹ peak power requirements (Fig. 3h). The latter computational estimates include aerodynamic added mass effects during wing motion and are detailed in the Supplementary Information.

At low Reynolds numbers, impermeable membranous wings barely outperform leaky bristled wings in generating aerodynamic force. Thus, the small advantage of using a membranous wing is overweighed by the advantage gained in reducing inertial torques and power by minimizing wing mass. This trade-off of energy savings for a small penalty in aerodynamic force generation is available only at Reynolds numbers of about 10 or lower, where sufficiently low leakiness can be achieved with a small number of slender bristles.

## Conclusions and outlook

The findings reported here expand our understanding of the flight mechanics at low Reynolds numbers. In flight, small insects need to produce forces to support their body weight in conditions of high viscous drag on the body and wings. *P. placentis* uses kinematic strategies that maximize wing flapping amplitude but at the potential cost of an increase in inertial power requirements. This is resolved by ptiloptery, an effective structural architecture that serves to reduce inertial costs of wing flapping, making elastic energy storage obsolete and reducing peak mechanical power requirements of the flight muscles. The wing-beat cycle of *P. placentis* is highly functionally divided into power and

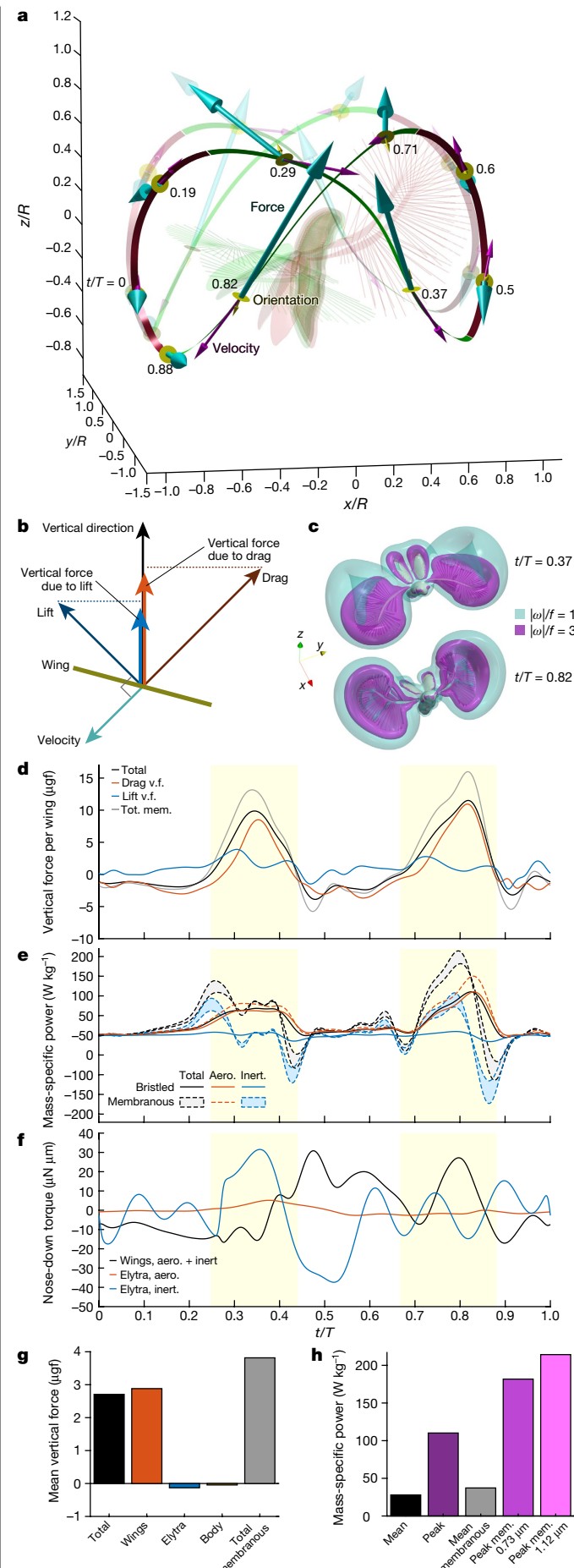

**Fig. 3 | Aerodynamic forces acting on the wings of *P. placentis*. a**, Wing tip trajectories and direction of total vertical force: downward force (recovery stroke) is shown in green, upward force (power stroke) is shown in red. Posture at $t/T$ of 0.6 is shown in red, and posture at $t/T$ of 0.82 is in green. Cyan arrows show aerodynamic force; magenta arrows show wing-tip velocity; yellow discs and arrows show dorsal surface orientation of the wing at nine labelled time instants. Opaque and transparent lines and arrows correspond to right and left wing, respectively. **b**, Vector scheme of forces acting on wing. **c**, Airflow simulation visualized using iso-surfaces of vorticity magnitude (see also Supplementary Video 5). **d**, Vertical aerodynamic force (v.f.) exerted on one wing versus time. Yellow highlighted zones denote the time span of power strokes. Tot. mem., vertical force of membranous wing model. **e**, Body mass-specific aerodynamic (aero.) and inertial (inert.) power, and their sum as the total power. **f**, Pitching torque about centre of mass. The positive direction is nose down. **g**, Contribution of different parts to total aerodynamic force acting on the beetle in the vertical direction, averaged over the wingbeat cycle. **h**, Mean and peak body mass-specific aerodynamic power in computations for bristled and membranous (mem.) wings.

slow-recovery strokes. The wings thereby produce pronounced high torques that cause the high-amplitude body pitch oscillation. Inertial braking provided by moving elytra represents an ingenious solution to this problem, enhancing posture stability without providing additional forces for flight. In *P. placentis*, these mechanisms improve the temporal distribution of muscle mechanical power requirements and help to maintain aerial performance at an extremely small body size. If this flight style is common for miniature beetles, it may largely explain their worldwide abundance. Further studies of other microinsects with bristled wings will help to reveal the causes of the convergent evolution of ptiloptery during miniaturization in many groups of insects.

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

## Methods

### Data reporting

No statistical methods were used to predetermine sample sizes. The experiments were not randomized and the investigators were not blinded to allocation during experiments and outcome assessment.

### Material

Adult featherwing beetles (*P. placentis* (Coleoptera: Ptiliidae)) were collected in Cát Tiên National Park, Vietnam, in November 2017. The beetles were collected and delivered to the laboratory together with the substrate for their safety. High-speed video recordings were made on the same day during a few hours after collecting.

### Morphology and morphometry

The material for morphological studies was fixed in alcoholic Bouin solution or in 70% ethanol. Wing structure was studied using a scanning electron microscope (SEM Jeol JSM-6380 and FEI Inspect F50), after dehydration of the samples and critical point drying, followed by gold sputtering. A confocal microscope (CLSM Olympus FV10i-O) and a transmitted light microscope (Olympus BX43) were also used, for which the samples were clarified and microscopic slides were made[26] (Supplementary Information). Measurements were taken from digital photographs in Autodesk AutoCAD software in ten replications (unless otherwise noted). Body weights and weights of particular body parts were calculated on the basis of three-dimensional reconstructions (Supplementary Information).

### Wing mass and moments of inertia

The volumes of the petiole and membranous part (the blade) of the wing were measured using CLSM image-based geometrical models. Uniform cuticle density 1,200 kg m$^{-3}$ was assumed[27]. The wing mass was obtained by summing up the contributions from the petiole, blade and setae. To calculate the mass of the setae, we first estimated their linear density (0.96 µg m$^{-1}$) using a three-dimensional model[25] and multiplied it by the length. The petiole and the blade of the wing model have constant thickness without veins. A possible range of the membrane thickness was hypothesized on the basis of measurements in *T. telengai* (Hymenoptera: Trichogrammatidae, body length 0.45 mm), *O. atomus* (Coleoptera: Corylophidae, body length 0.8 mm) and *L. atomus* (Coleoptera: Hydraenidae, body length 1.1 mm), on 0.5 µm thick histological sections obtained by diamond knife cutting using a Leica microtome, after fixation and embedding in araldite. These values are the minimal thicknesses measured in each species. The measurements were performed using an Olympus BX43 microscope. The measurement error of linear dimensions is of the order of magnitude of 1% in the spanwise and chordwise directions and 10% for the thickness. The s.d. of wing cuticle density[25] is approximately 100 kg m$^{-3}$. This suggests that the overall root sum square error of the wing mass calculation is of about 13%. To evaluate the moments of inertia, surface density of the membranous parts and linear density of the bristles were calculated. The moments of inertia of the individual setae were calculated using the formula for a thin rod at an angle and the parallel axis theorem. The moments of inertia of the membranous parts were calculated using a two-dimensional quadrature rule with the discretization step of 50 µm.

### High-speed recording

Flight of the beetles was recorded in closed 20 × 20 × 20 mm chambers, custom made of 1.0 mm thick microscopic slides and 0.15 mm cover-glass at a natural level of illumination in visible light. There were 20–30 insects in the flight chamber during the recording. For temperature stabilization the flight chamber was chilled by an air fan from the outside. The ambient temperature measured by a digital thermocouple was 22–24 °C; the temperature of the flight chamber was 22–26 °C.

High-speed video recordings were made using two synchronized Evercam 4000 cameras (Evercam) with a frequency of 3,845 FPS and a shutter speed of 20 µs in infrared light (850 nm LED). The high-speed cameras were mounted on optical rails precisely orthogonal to each other and both situated at 0° from the horizon. Two IR LED lights were placed opposite to the cameras and one light above the flight chamber. A graphical representation of the experimental setup can be found in the previous study[2].

### Measurement of kinematics

For analysis, 13 recordings were selected. For four of them (PP2, PP4, PP5 and PP12) we reconstructed the kinematics of body parts in four kinematic cycles for each and performed CFD calculations because the flight of these specimens was especially similar to conventional hovering: relatively slow normal flight with horizontal velocity 0.057 ± 0.014 m s$^{-1}$ (hereafter mean ± s.d.) and 0.039 ± 0.031 m s$^{-1}$ vertical velocity (PP2, PP4, PP5 and PP12). In CFD analysis with the membranous wing model, we selected kinematics of PP2, which does not cross the wings while clapping. This case is convenient for comparing the performance of bristled wings with substitute membranous wings, because it guarantees that the latter do not intersect. The perimeter of the membrane is formed by lines connecting the tips of the bristles (see the previous study[25] for more information). The descriptions of kinematics and aerodynamics, as well as the illustrations, refer to results obtained for individual PP2. For the results obtained for other specimens, see Supplementary Information and Extended Data Figs. 2, 4–6.

Average wingbeat frequency was calculated as the mean of the wingbeat frequency in all recordings. In each recording, the number of frames was counted in several complete kinematic cycles, 104 cycles in total.

For the mathematical description of the kinematics of the wings and elytra, we used the Euler angles system[28,29] (Fig. 2b) based on frame-by-frame reconstruction of the location of the insect's body parts (wings, elytra and body itself) performed in Autodesk 3Ds Max. Three-dimensional models of the body and elytra were obtained by confocal microscope image stacking, and the flat wing model was based on light microscopy photos of dissected wings. We used the rigid flat wing model for reconstruction of the kinematics because the deformations of the wings are minor (Supplementary Information). First, we prepared frame sequences with four full kinematical cycles in each. The frames were then centred and cropped by point between the bases of the wings and then placed as orthogonal projections. Virtual models of body parts were placed into a coordinate system with two image planes. Then we manually changed the position and rotated body parts until their orthogonal projections were superimposed on the image planes. For calculating the Euler angles, a coordinate system was created (Fig. 2a). The *XOY* plane is a plane parallel to the stroke plane, and intersecting with the base of wing or elytron, which is positioned in the zero point. To determine the position of the stroke plane, we calculated the major axis trend line of the wingtip coordinates instead of the linear trend line[29], because the wingtip trajectory of *P. placentis* forms a wide scatter plot. Stroke deviation angle ($\theta$) and positional angle ($\varphi$) were calculated from the coordinates of the base and apex. Pitch angle ($\psi$) is the angle between the stroke plane and the chord perpendicular to the line between the base and apex. The body pitch angle ($\chi$) is the angle between the stroke plane and longitudinal axis of the body, calculated as the line between the tip of the abdomen and the midpoint between the apical antennomeres. Pitch angle ($\beta$) of the stroke plane relative to the horizon was also measured.

For flight speed analysis we performed tracking of the centre of the body (middle point between the extreme edges of the head and abdomen) in Tracker (Open Source Physics) in both projections and calculated the instantaneous velocity and its vertical and horizontal components in each frame. The obtained speed values were filtered by loess fitting in R (stats package). The minimum distance between the wingblade tips during bottom claps was also calculated.

## Computational fluid dynamics

Time intervals of low-speed flight with duration longer than four wing beats were selected. The angles $\varphi$, $\theta$ and $\psi$ of the left wing, right wing and elytra and the body angle $\chi$ were interpolated on a uniform grid with time step size $\Delta t = 2.6 \times 10^{-6}$ s. By solving numerically $\varphi(t) = 0$ with respect to $t$, we identified four subsequent wingbeat cycles and calculated the average cycle period $T$ and the wingbeat frequency $f = 1/T$. We then spline-interpolated the data for each of the four cycles on a grid subdividing the time interval $[0, T]$ with step $\Delta t$, calculated phase averages, then calculated the average between the left and right wing. This yielded the plots shown in Fig. 2c, d. Constant forward and upward/downward flight velocity was prescribed using the time average values of the loess-filtered time series.

The computational fluid dynamics analysis was performed using the open-source Navier–Stokes solver WABBIT[30], which is based on the artificial compressibility method to enforce velocity-pressure coupling, volume penalization method to model the no-slip condition at the solid surfaces, and dynamic grid adaptation using the wavelet coefficients as refinement indicators. The flying insect was represented as an assembly of five rigid solid moving parts: the two elytra and the two wings move relative to the body, and the body oscillates about its lateral axis (Supplementary Information). The kinematic protocol is described in Supplementary Information and Extended Data Fig. 2c. The computational domain is a $12R \times 12R \times 12R$ cube, where $R$ is the wing length, with volume penalization used in combination with periodic external boundary conditions to enforce the desired far-field velocity[30]. The computational domain was decomposed in nested Cartesian blocks, each containing $25 \times 25 \times 25$ grid points. The blocks were created, removed and redistributed among parallel computation processes so as to ensure maximum refinement level near the solid boundaries and constant wavelet coefficient thresholding otherwise during the simulations. The numerical simulations started from the quiescent air condition, continued for a time period of two wingbeat cycles with a coarse spatial grid resolution of $\Delta x_{min} = 0.00781R$ to let the flow develop to its ultimate periodic state, then the spatial discretization size was allowed to reduce to $\Delta x_{min} = 0.00098R$ if the wing was bristled or to $\Delta x_{min} = 0.00049R$ if it was membranous, and the simulation continued for one more wingbeat period to obtain high-resolution results. The air temperature was 25 °C in all cases; its density was $\rho = 1.197$ kg m$^{-3}$ and its kinematic viscosity was $v = 1.54 \times 10^{-5}$ m$^2$ s$^{-1}$; the artificial speed of sound was prescribed as $c_0 = 30.38fR$, based on an earlier experimental validation[25]. The volume penalization and other case-specific parameter values are provided in Supplementary Information. The CFD simulation accuracy is discussed in Supplementary Information and Extended Data Fig. 8.

## Decomposition of the aerodynamic force of a wing into lift and drag components

The drag component of the total instantaneous aerodynamic force acting on the wing is defined as its projection on the direction of the wing velocity at the radius of gyration. The lift component is defined as a vector subtraction of the total force and the drag component. The total lift and drag force vectors are projected on the vertical ($z$) direction to obtain the time courses shown in Fig. 3d.

## Reporting summary

Further information on research design is available in the Nature Research Reporting Summary linked to this paper.

## Data availability

Extended data sets and raw data are available in an Open Science Framework repository (https://osf.io/v3wrk/).

## Code availability

Computational fluid dynamics simulations were performed using an open-source code WABBIT, which can be downloaded from GitHub (https://github.com/adaptive-cfd/WABBIT) and has been described in detail elsewhere[30].

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

**Acknowledgements** A.A.P. thanks R. D. Zhantiev for mentorship and for inspiring him to study miniature insects. We thank A. K. Tsaturyan for helpful discussions. The work of S.E.F., A.A.P., P.N.P. and N.A.L. was supported by the Russian Science Foundation (project number 19-14-00045, study of morphology, high-speed recording and reconstruction of kinematics). This study was performed using equipment of the shared research facilities of HPC computing resources at Lomonosov Moscow State University (A.A.P. and D.K., project no. 2183 'Computational fluid dynamics of the smallest insects'), TSUBAME3.0 supercomputer at the Tokyo Institute of Technology (D.K.) and HPC resources of IDRIS (T.E., allocation number A0102A01664 attributed by the Grand Équipement National de Calcul Intensif (GENCI)). The work of D.K. was supported by the JSPS KAKENHI (grant number 18K13693). H.L. was partially supported by the JSPS KAKENHI (grant number 19H02060). The contributions of F.-O.L. and T.E. were supported by grants from the Deutsche Forschungsgemeinschaft to F.-O.L. (LE905/16-1 and LE905/18-1). SEM studies were performed using the Shared Research Facility Electron microscopy in life sciences at Lomonosov Moscow State University (unique equipment 'Three-dimensional electron microscopy and spectroscopy').

**Author contributions** A.A.P. conceptualized and designed this study; S.E.F., N.A.L and A.A.P. designed the experiment and collected the data; H.L., F.-O.L and R.O. conceptualized the computational analysis; T.E. and D.K. performed the CFD simulations; S.E.F. and D.K. analysed the data; S.E.F., P.N.P., D.K. and A.A.P. wrote the manuscript. All authors edited the manuscript and approved the final version.

**Competing interests** The authors declare no competing interests.

**Additional information**
**Correspondence and requests for materials** should be addressed to Sergey E. Farisenkov or Alexey A. Polilov.

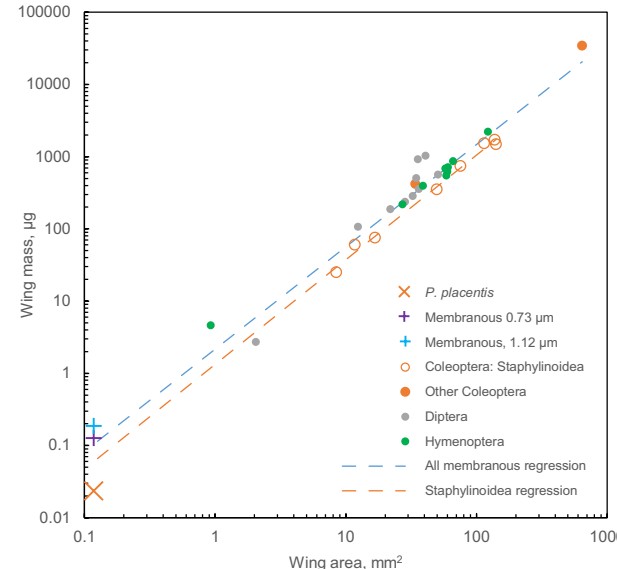

**Extended Data Fig. 1 | Relationship between wing mass $m_w$ and wing area $S$.**
Filled circles show data points for living mass of membranous insect wings
from published sources (see Supplementary Information). Punctured circles
show dried mass of membranous wings of Staphylinoidea beetles obtained as
described in Supplementary Information. The cross indicates calculated
weight of the bristled wing of the featherwing beetle *Paratuposa placentis*.
Plus signs delimit the estimated mass range of equivalent membranous wings.
Blue dashed line is an allometric trend based on all data for membranous wings
($m_w = 2.17 \cdot S^{1.42}$, $R^2 = 0.946$). Orange dashed line is an allometric trend based on
data for dried wings of Staphylinoidea ($m_w = 1.33 \cdot S^{1.45}$, $R^2 = 0.992$). The data
correspond to mass and area of one wing in Coleoptera and Diptera and of one
forewing-hindwing pair in Hymenoptera.

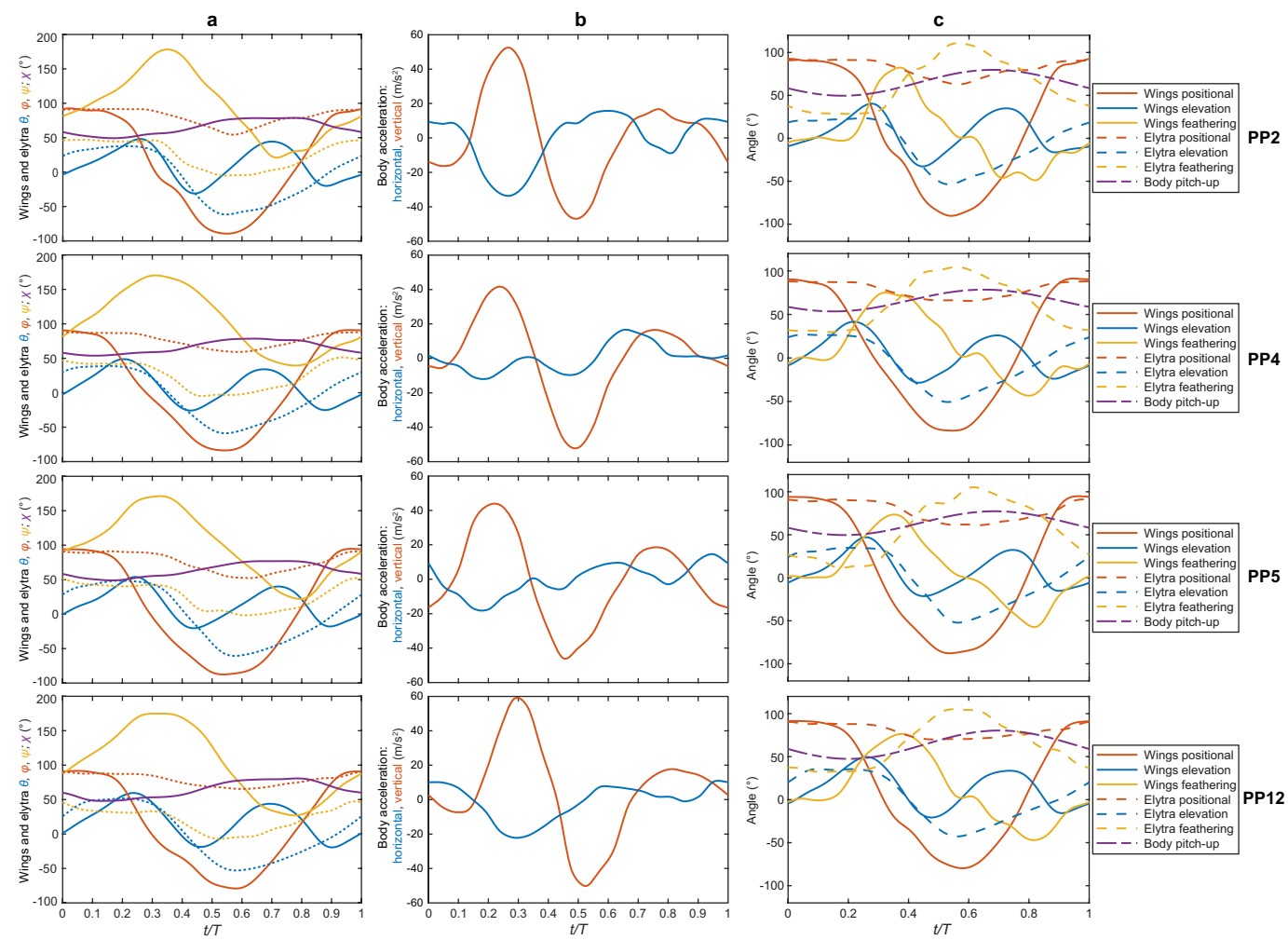

**Extended Data Fig. 2 | Kinematics descriptions of *Paratuposa placentis* individuals PP2, PP4, PP5, PP12. a**, Euler angles of wings and elytra and body pitch angle. **b**, Vertical and horizontal components of body acceleration. **c**, Euler angles of wings and elytra and body pitch-up angle between horizontal plane and longitudinal axis of body, prepared for CFD.

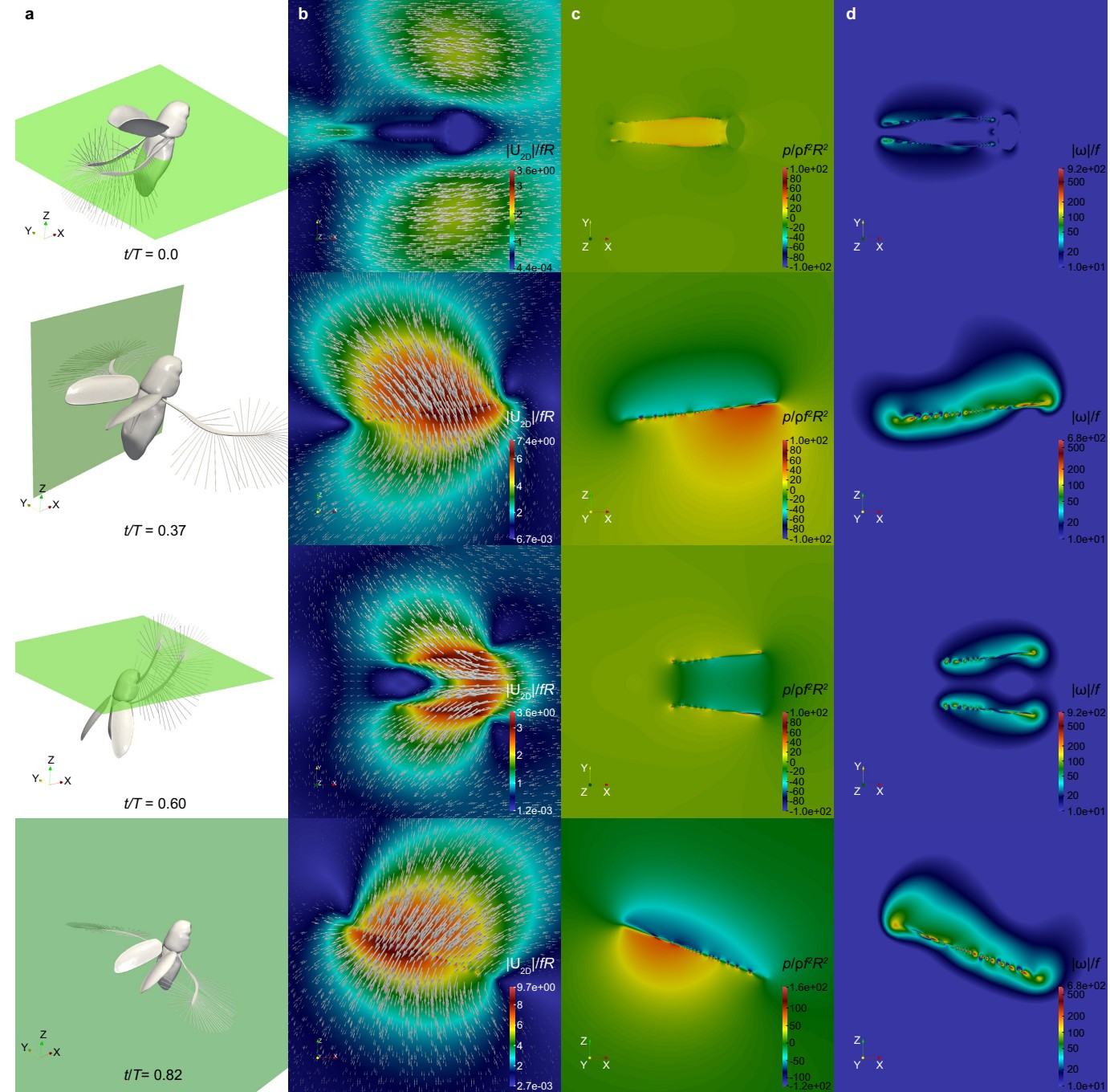

**Extended data Fig. 3 | CFD simulation of the flight of *Paratuposa placentis*: visualization of the flow using two-dimensional slices in the 3rd simulation cycle.** Time instances are $t/T = 0.0$ (dorsal recovery stroke) $t/T = 0.37$ (first power stroke), $t/T = 0.60$ (ventral recovery stroke) and $t/T = 0.82$ (second power stroke), from top to bottom. **a**, Planes perpendicular to wings.

**b**, Velocity vectors (arrows) and velocity magnitude (background colouring), showing two-component in-plane velocity with the third vector component omitted for clarity. **c**, Pressure. **d**, Magnitude of three-dimensional vorticity with colouring in logarithmic scale.

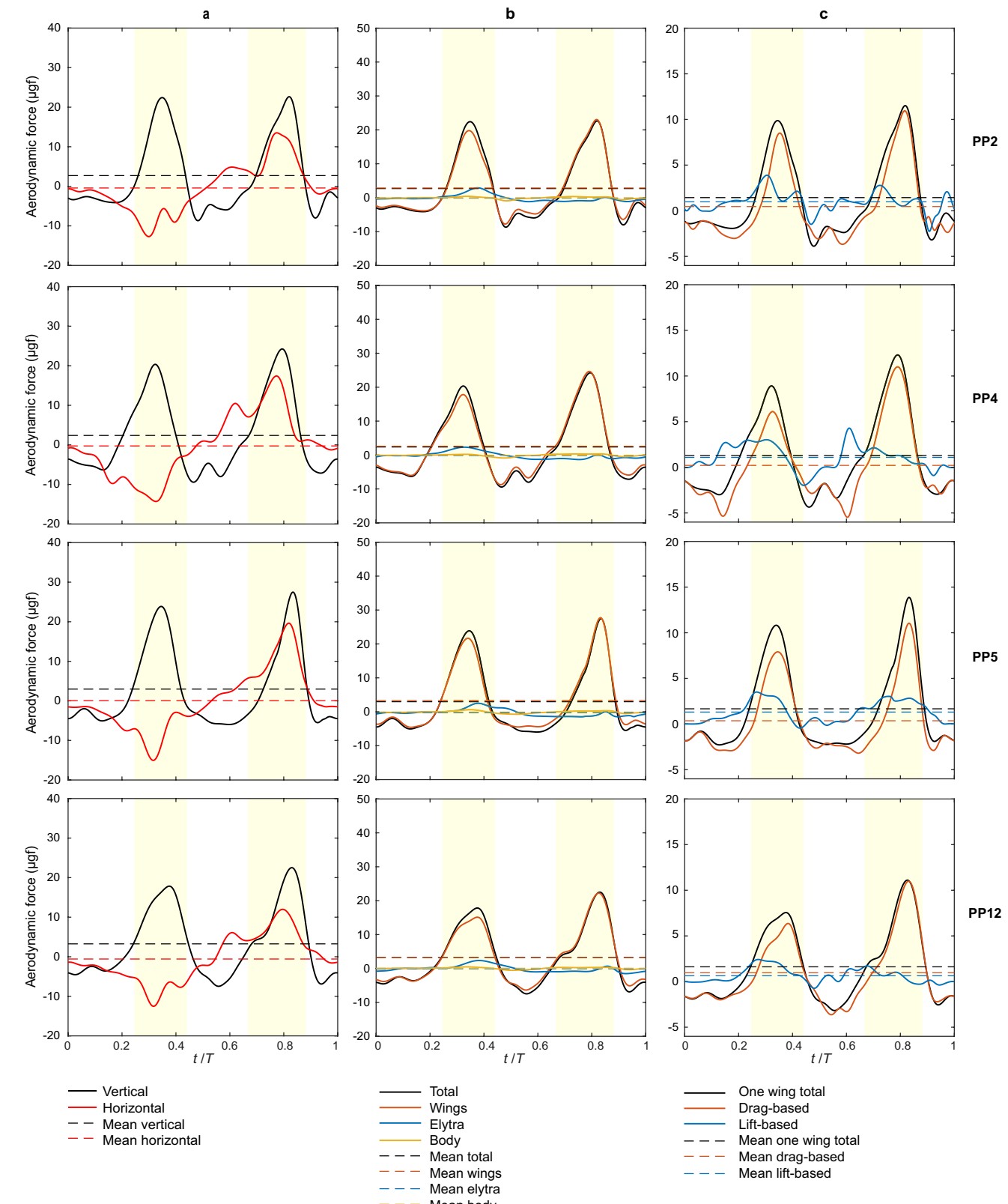

**Extended Data Fig. 4 | Calculated components of aerodynamic force of** ***Paratuposa placentis*** **individuals PP2, PP4, PP5, PP12. a**, Vertical and horizontal components of the aerodynamic force exerted on the insect. **b**, Total vertical aerodynamic force acting on the insect and its breakdown into the vertical component of the aerodynamic force on the pair of wings, the pair of elytra and the body. **c**, Lift-drag decomposition of the vertical force acting on one wing.

**a**

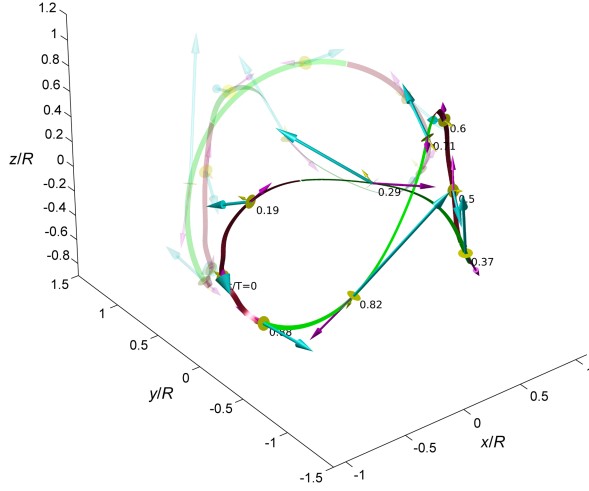

**b**

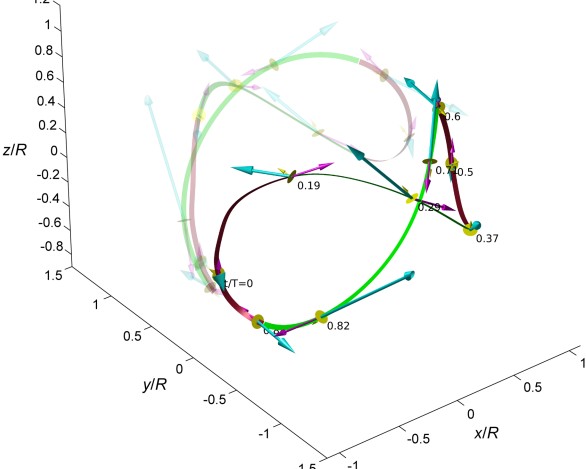

**c**

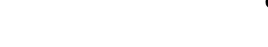
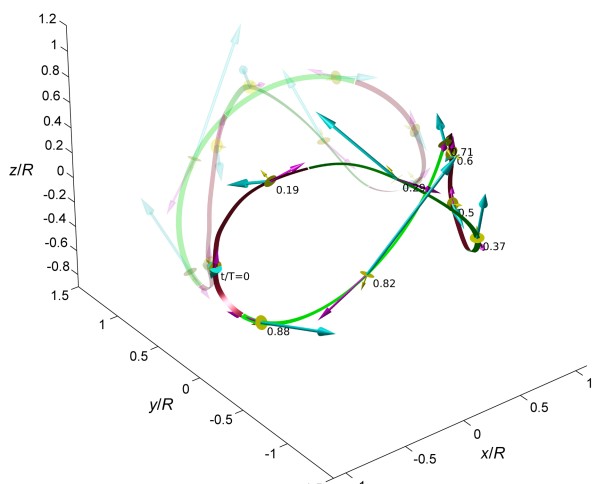

**d**

**Extended Data Fig. 5 | Three-dimensional reconstruction of wing-tip trajectories (continuous lines), aerodynamic force vectors (cyan arrows), velocity vectors (magenta arrows) and wing orientation (yellow circles and arrows) during flight in *Paratuposa placentis* individuals. a**, PP2. **b**, PP4. **c**, PP5. **d**, PP12. For interactive 3D pdf version, see Supplementary Fig. 9.

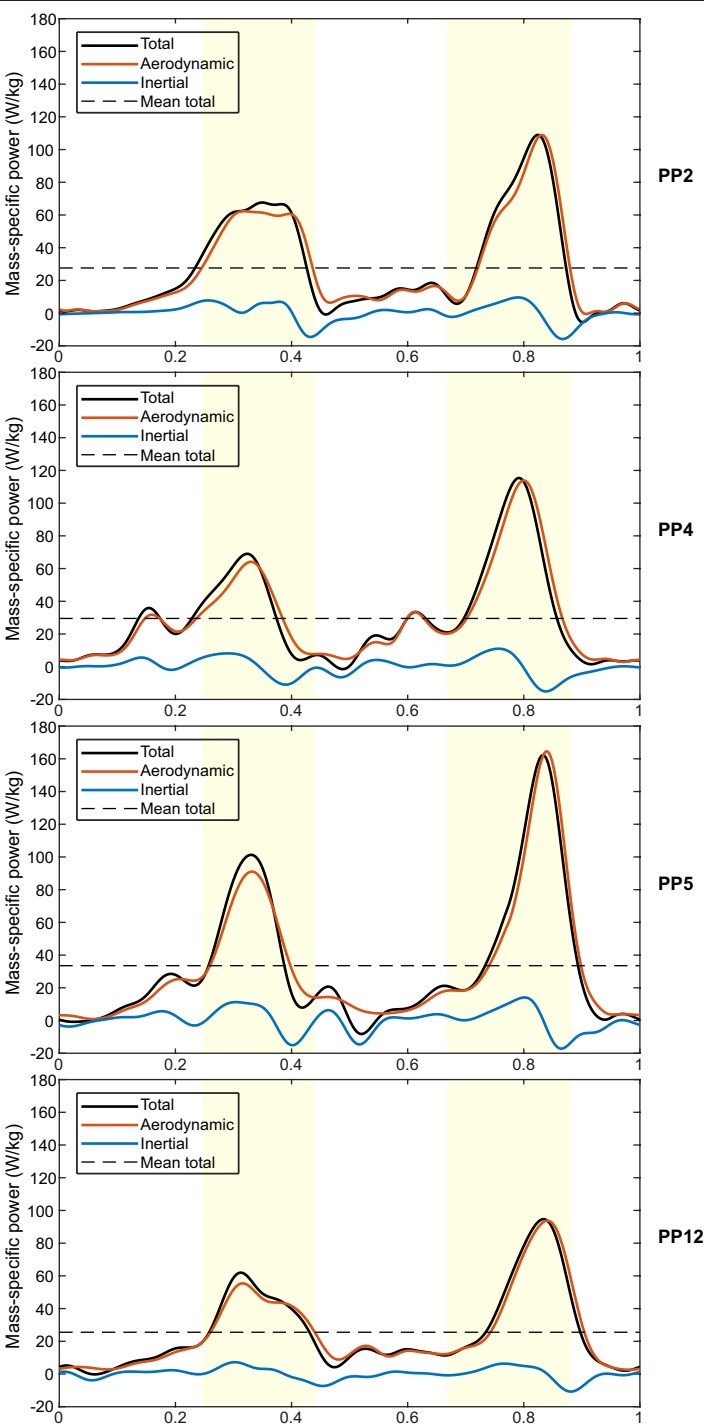

**Extended Data Fig. 6 | Body-mass-specific mechanical power components of *Paratuposa placentis* individuals PP2, PP4, PP5 and PP12.** The mechanical power essentially remains positive through the entire wingbeat cycle period, although occasionally it takes small negative values. The aerodynamic contribution is dominant.

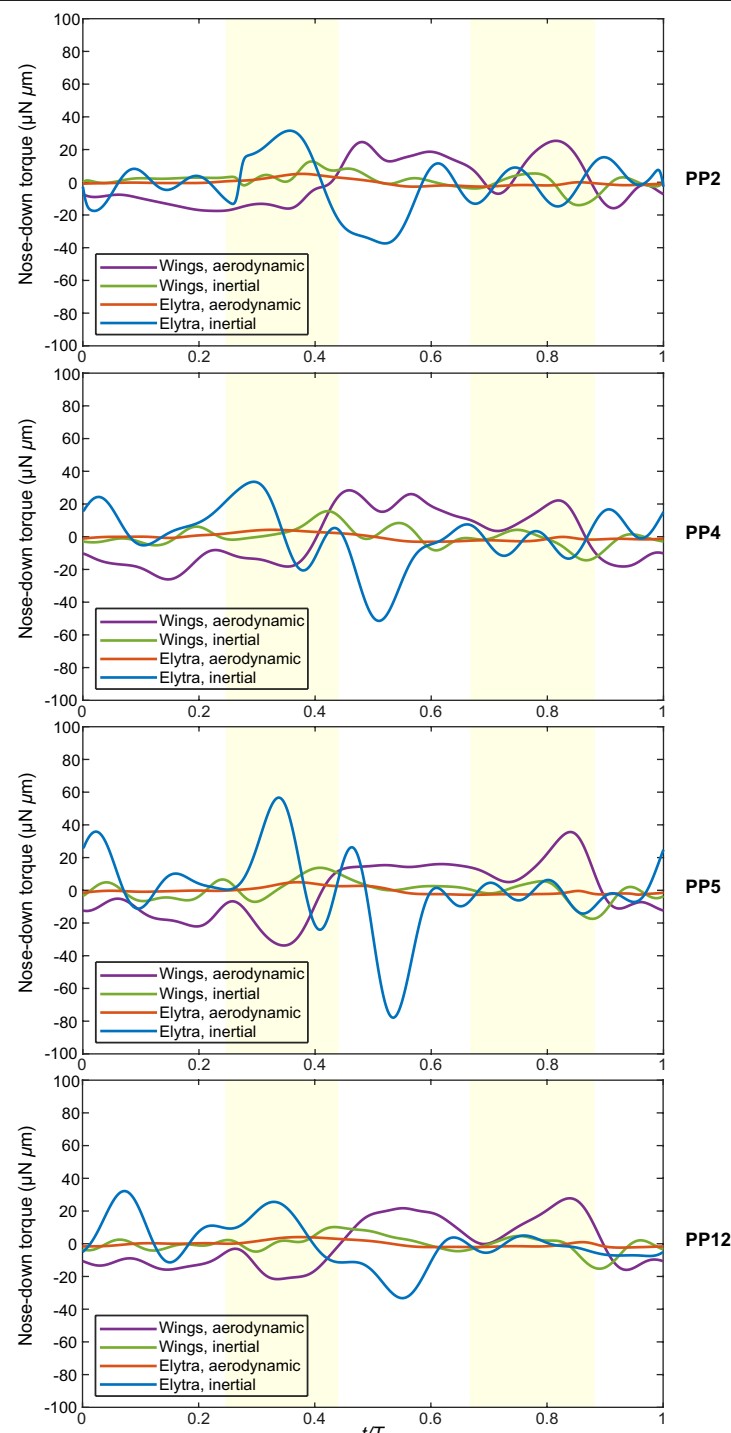

**Extended Data Fig. 7 | Components of pitching moment acting on *Paratuposa placentis* individuals PP2, PP4, PP5, PP12.** Positive direction is nose down.

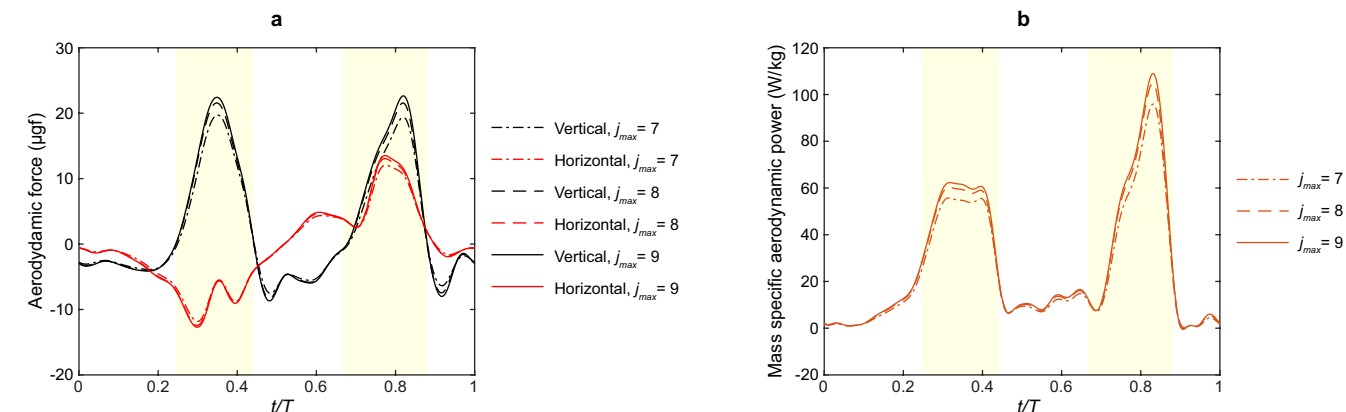

**Extended Data Fig. 8 | Aerodynamic performance of *Paratuposa placentis* individual PP2 obtained from three different simulations using three different values of maximum refinement level $j_{max}$. a**, Aerodynamic force. **b**, Mass-specific aerodynamic power.

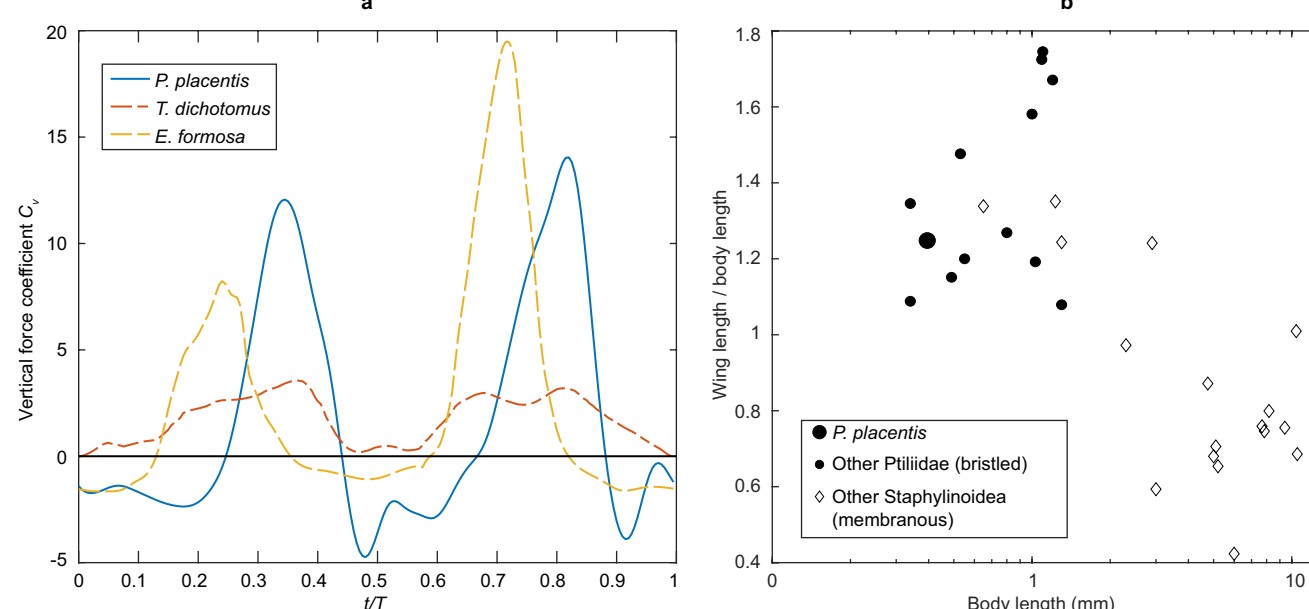

**a**

**b**

**Extended Data Fig. 9 | Comparison of *Paratuposa placentis* and other insects that have bristled or membranous wings. a**, Time variation of the vertical force coefficient in three different species. Data for the large rhinoceros beetle *Trypoxylus dichotomus* and for the tiny chalcid wasp *Encarsia formosa* are adapted so that the cycle begins with the downstroke. The force coefficient is defined as $C_V = 2F_V/\rho(2\Phi R_g f)^2 S$, where $F_V$ is vertical force, $\rho$ is air density, $\Phi$ is flapping amplitude, $R_g$ is wing geometric radius of gyration, $f$ is flapping frequency, and $S$ is wing area. Note that the bristled wings of *Encarsia formosa* were modelled as impermeable solid plates. For additional information, see Supplementary Information. **b**, Wing length relative to body length.

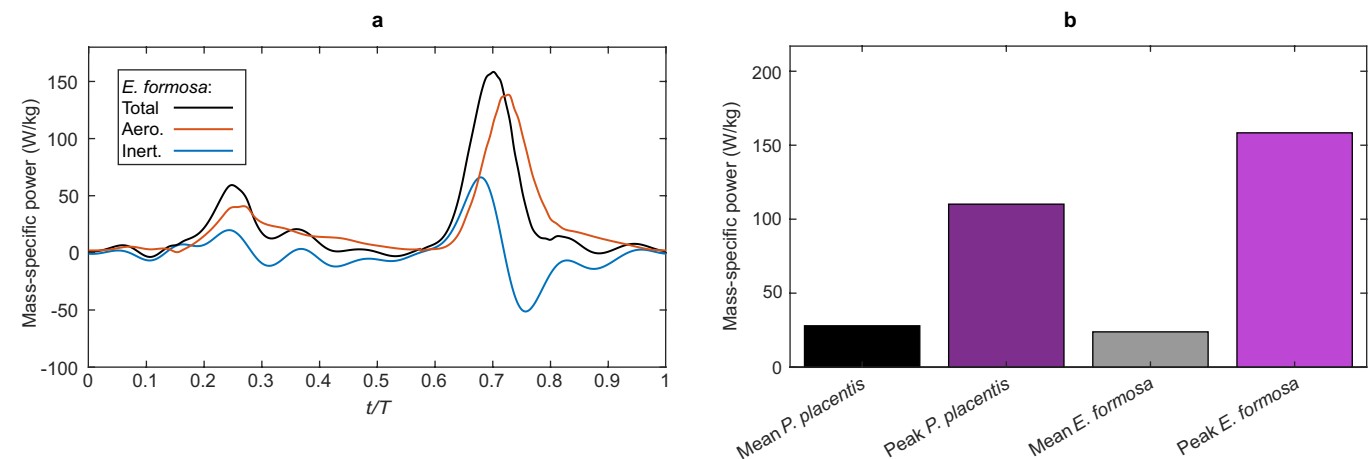

**Extended Data Fig. 10 | Comparison of *Paratuposa placentis* and *Encarsia formosa*. a**, Time variation of body-mass-specific mechanical power. **b**, Mean (averaged over wingbeat cycle) and peak (maximum over the cycle) body-mass-specific mechanical power.

Sergey Farisenkov

# Reporting Summary

Nature Research wishes to improve the reproducibility of the work that we publish. This form provides structure for consistency and transparency in reporting. For further information on Nature Research policies, see our Editorial Policies and the Editorial Policy Checklist.

## Statistics

For all statistical analyses, confirm that the following items are present in the figure legend, table legend, main text, or Methods section.

| n/a | Confirmed | |
|---|---|---|
| ☐ | ☒ | The exact sample size (*n*) for each experimental group/condition, given as a discrete number and unit of measurement |
| ☐ | ☒ | A statement on whether measurements were taken from distinct samples or whether the same sample was measured repeatedly |
| ☐ | ☒ | The statistical test(s) used AND whether they are one- or two-sided<br>*Only common tests should be described solely by name; describe more complex techniques in the Methods section.* |
| ☐ | ☒ | A description of all covariates tested |
| ☐ | ☒ | A description of any assumptions or corrections, such as tests of normality and adjustment for multiple comparisons |
| ☐ | ☒ | A full description of the statistical parameters including central tendency (e.g. means) or other basic estimates (e.g. regression coefficient) AND variation (e.g. standard deviation) or associated estimates of uncertainty (e.g. confidence intervals) |
| ☐ | ☒ | For null hypothesis testing, the test statistic (e.g. *F*, *t*, *r*) with confidence intervals, effect sizes, degrees of freedom and *P* value noted<br>*Give P values as exact values whenever suitable.* |
| ☒ | ☐ | For Bayesian analysis, information on the choice of priors and Markov chain Monte Carlo settings |
| ☒ | ☐ | For hierarchical and complex designs, identification of the appropriate level for tests and full reporting of outcomes |
| ☒ | ☐ | Estimates of effect sizes (e.g. Cohen's *d*, Pearson's *r*), indicating how they were calculated |

*Our web collection on statistics for biologists contains articles on many of the points above.*

## Software and code

Policy information about availability of computer code

| Data collection | The high-speed recordings were filmed in Evercam SRV-HS. Microphotographs were received in Tucsen Mosaic 2.0. 3D reconstruction of body parts were performed in Bitplane Imaris v.9.5.<br>Non-standard or custom-designed programmes or codes were not used. |
|---|---|
| Data analysis | Primary video processing was performed using ImageJ v.1.52p. Morphometric analysis were performed using Autodesk AutoCAD 2015.Kinematic data processing were performed using Autodesk 3ds Max 15.0 and the standard packages of Matlab R2019b v.3.1 and open-source R packages in R v.4.0.0.28 enviroment. Computational fluid dynamics simulations were performed using an open-source code WABBIT. It can be downloaded from GitHub (https://github.com/adaptive-cfd/WABBIT) and it has been described in detail elsewhere [30]. |

For manuscripts utilizing custom algorithms or software that are central to the research but not yet described in published literature, software must be made available to editors and reviewers. We strongly encourage code deposition in a community repository (e.g. GitHub). See the Nature Research guidelines for submitting code & software for further information.

## Data

Policy information about availability of data

All manuscripts must include a data availability statement. This statement should provide the following information, where applicable:

- Accession codes, unique identifiers, or web links for publicly available datasets
- A list of figures that have associated raw data
- A description of any restrictions on data availability

Extended data sets and faw data are available in the following Open Science Framework repository: https://osf.io/v3wrk/

# Field-specific reporting

Please select the one below that is the best fit for your research. If you are not sure, read the appropriate sections before making your selection.

☒ Life sciences          ☐ Behavioural & social sciences          ☐ Ecological, evolutionary & environmental sciences

For a reference copy of the document with all sections, see nature.com/documents/nr-reporting-summary-flat.pdf

# Life sciences study design

All studies must disclose on these points even when the disclosure is negative.

| | |
|---|---|
| Sample size | Sample sizes for kinematic reconstruction and analysis, and CFD analysis were determined by quantity and quality of high-speed videos available. The complete reconstruction of the kinematics and CFD was done for four individual beetles, four wing beat cycles of each beetle. The other characteristics of the kinematics were measured using all the 13 recordings or some of them. Each recording contains 3 to 23 full wing beats.<br>The sample sizes for morphometrics comprised at least 10 replications. Statistical methods were not used to predetermine the sample size. Sample sizes were selected based on the limited amount of material available. |
| Data exclusions | No data were excluded. |
| Replication | Experiments were performed independently. |
| Randomization | Artificial randomization is not relevant to this study because the original video recordings were obtained from random samples, and then all of them were analysed. |
| Blinding | Blinding was not possible because the analysis of the data was carried out by the persons involved in the data collection, and the interpretation of the data was carried out by the persons responsible for the analysis. |

# Reporting for specific materials, systems and methods

We require information from authors about some types of materials, experimental systems and methods used in many studies. Here, indicate whether each material, system or method listed is relevant to your study. If you are not sure if a list item applies to your research, read the appropriate section before selecting a response.

### Materials & experimental systems

| n/a | Involved in the study |
|---|---|
| ☒ | ☐ Antibodies |
| ☒ | ☐ Eukaryotic cell lines |
| ☒ | ☐ Palaeontology and archaeology |
| ☐ | ☒ Animals and other organisms |
| ☒ | ☐ Human research participants |
| ☒ | ☐ Clinical data |
| ☒ | ☐ Dual use research of concern |

### Methods

| n/a | Involved in the study |
|---|---|
| ☒ | ☐ ChIP-seq |
| ☒ | ☐ Flow cytometry |
| ☒ | ☐ MRI-based neuroimaging |

## Animals and other organisms

Policy information about studies involving animals; ARRIVE guidelines recommended for reporting animal research

| | |
|---|---|
| Laboratory animals | The study did not involve laboratory animals. |
| Wild animals | The featherwing beetle Paratuposa placentis (Coleoptrea: Ptiliidae) is a widespread and abundant species. It is not a protected species, and working with it does not require special permits. The sex of the beetles was not determined, because this species has no pronounced sexual dimorphism, and the sex of an individual beetle cannot be determined without dissecting, which would have been incompatible with most of the methods used. The study was performed on adult beetles; their exact age could not be determined, because the beetles were taken from nature, and no characters that can be used to determine the age of adult beetles of this species are known.<br>The beetles were captured in the natural habitat with the substrate and brought to the laboratory in containers for recording their flight. After the video recording session, most beetles were released in the collection locality in the same day, and only the minimum required number of specimens for the morphological part of the study The insects were anesthetized with $CO_2$ and then fixed in Bouin's splution or 70% ethanol according to the standard method. |
| Field-collected samples | The beetles were kept in ventilated containers with the substrate for several hours after capture in laboratory (temperature 22-24 degree, natural lighting conditions) and were released or fixed after experiments. |

Ethics oversight | Ethics oversight was not required because ptiliid beetles do not require IRB or ethics approval. The studied beetles do not belong to protected or endangered species.
Field work was carried out in the framework agreement № 37/HD on the scientific cooperation between Cat Tien National Park and the Joint Russian–Vietnamese Tropical Research and Technological Centre.

Note that full information on the approval of the study protocol must also be provided in the manuscript.

