## [Peer Review File · Nature]

Manuscript Title: Novel flight style and light wings boost flight performance of tiny beetles

Editorial Note:

Redactions – Third Party Material

Reviewer Comments & Author Rebuttals

Reviewer Reports on the Initial Version:

Referee #1 (Remarks to the Author):

This paper is a tour-de-force combination of 3D kinematics obtained from multi-camera high-speed-video of free-flying Paratuposa beetles together with computational fluid dynamics modelling to analyse the flight of tiny feather-winged beetles. The technical challenge in just obtaining the video is immense, the analysis is ground-breaking and the results are fascinating. I recommend publication. I have only a few minor suggestions relating to power calculations.

The authors show by calculation that the light weight of the feather wings compared to normal membranous wings reduces inertial costs of flapping. The calculations on power requirements are expressed as watts per kilo of bodyweight. I am deeply impressed that Paratuposa manages to fly at 28W/kg average (lines 161-1740. Professional cyclists manage about 6W/kg (the tour-de-france is imminent), the comparison might be worth making for general readers, and a comparison with other flying animals would be useful to put these numbers in context. However, what I would really like to know is the power output of the beetle's muscles required for flight. Peak muscle power output is around 200W/kg (aerobic), that seems likely to be challenged by these beetles - which would further emphasise the author's correct claim that feathered wings are necessary for flight at such small sizes to minimise inertia. What is the average and instantaneous peak power output of the muscles, in W/kg? It must be possible to estimate muscle mass, or at least thoracic mass, from the SEM images.

More could be made of the fact that at $Re < 10$ viscosity dominates, and these beetles are flying in the same regime as tiny swimmers like daphnia and diving beetles which also use feathered propulsive.

Figure 2 A (the video sequence) is blurry and the supplementary video is much clearer, can anything be done to enlarge and clarify the images of the beetle wings? The clap is not readily discernible.

These points are minor. I recommend publication.

Referee #2 (Remarks to the Author):

In their manuscript, Farisenkov and coworkers present the results of a morphological, kinematic, and aerodynamic analysis of a ptiliid beetles, which are among the tiniest of all flying insects. The Ptiliidae are known for the feather-like wings that the smallest members of the group possess. The research team has managed to collect high speed video of sufficient quality to perform a kinematic and aerodynamic analysis, thereby providing the first quantitative look at the flight biomechanics of this noteworthy group of small insects. While feather wings are not unique to the Ptiliidae, the

results of this manuscript have the potential to answer interesting questions regarding the evolution of this morphological specialization that certainly finds an extreme within this group.

The paper makes two major claims about the flight biomechanics of *Paratuposa placentis*, the first is that it executes a so-called 'novel' flight style in which each half stroke execute a downward directed, drag-dominated power stroke followed by a rapid upward recovery in which the wing is rotated so as to produced less drag. The resulting full stroke produces two prominent upward force peaks per cycle. The second main claim is that due to the low mass of the wing, the inertial costs of flight (i.e. the cost to accelerate the wing mass back and forth) are extremely small, such that nearly all mechanical power cost goes into the production of aerodynamic power, and would so even in the absence of an elastic storage mechanism to reduce inertial power. In this view, ptiloptery evolved as a means of lowering flight costs by massively reducing wing mass—an option that is only available to insects operating at low Reynolds numbers where they can make use of the reduced leakiness of wings formed of arrays of fine hairs.

While I am impressed with the comprehensive nature of the manuscript, which relies on experts across an array of different disciplines and the fascinating look at the flight biology of these tiny creatures, I believe that the manuscript as it currently stands contains several critical flaws that precludes its publication in *Nature* at this time. I describe these issues below with the hope the authors can institute a substantive set of revisions that might render the paper more accurate and suitable for publication.

Major Concerns

1) The quality of writing is generally poor compared to what is expected for a publication in *Nature*. The paper is replete with critical claims and calculations (most critically, the calculations of wing mass) that are buried in the supplemental material rather than being presented transparently in the main text. Having read both the main text and supplemental material very carefully, I feel that the authors have done a poor job deciding what material belongs in the main text and what belongs in the supplement. This lack of clarity starts with the very title of the paper, 'A novel flight style allowing the smallest featherwing beetles to excel'. This title is vague and tautological—any animal that is not extinct could be said to 'excel'—and does not summarize the main claim in the paper, which is that feather wings evolved in the Ptiliidae to minimize flight cost due to a reduction in inertial power requirements. Another issue that appears throughout the paper is a rather flagrant disregard for proper use of the concept of significant digits within nearly all calculations. Many numbers are provided with decimal value precisions that are difficult to justify given the precision of the measured values or constants upon which they are based.

2) The title of the paper and text throughout implies that *P. placentis* uses a novel form of flight kinematics and aerodynamics, in which vertical force is generated by a downward plunge (at high aerodynamic angle of attack) at the start of each upstroke, followed by an upward recovery stroke (with low aerodynamic angle of attack). However, textbooks on animal flight are literally replete with sections entitled 'Drag-based propulsion' or 'Drag vs. Lift as a way to get thrust', as well as illustrations of birds and insects employing a figure-of-eight pattern of wing motion. The kinematics of *P. placentis* are extreme to be sure, but there is nothing unprecedented or unexpected in what they do compared to prior studies. Studies of hovering dragonflies have identified a drag-based mechanism in which the wing produces primarily drag during the downstroke and little reverse drag during the upstroke. Many authors have noted the U-shaped curve traced by the path of hovering insects (e.g. any one of the many published studies of *Drosophila*) and how the first portion of each downstroke and upstroke (more properly, forward stroke and backward stroke) contribute substantially to vertical force due to the downward motion and large aerodynamic angle of attack of the wing at that time. Most notably, Cheng and Sun published a paper this year (*Physics of Fluids* 33, 021905) on the kinematics and aerodynamics of *Encarsia formosa*, which execute a pronounced U-shaped pattern of wing motion that generates to prominent vertical force peaks at a Reynold number of ~ 10 (almost exactly the same Re as *P. placentis*). The describing those results, Cheng and Sun wrote, "Therefore, very small insects must

use a “new” flapping mode.” I would probably qualify their statement to say that very small insects use a more distorted form of an “old” flapping mode, but the point is that neither the kinematics described here for *P. placentis* nor the drag-based mechanism that result can be accurately be described as ‘novel’. The authors themselves have reproduced the force traces reported in a prior preprint of the Cheng and Sun paper in Figure 4. Surely, the fundamental similarity between the results garnered from *E. formosa* and *P. placentis* and are more striking and noteworthy than the small differences, which are just as likely to have emerged from subtle inaccuracies in kinematic measurements or CFD simulations than from any fundamental difference in aerodynamics. Put plainly, the pattern of motion identified in *P. placentis* is not fundamentally different (or unexpected) from what has been recorded from a small insect using primarily solid wings. Further, as the authors note, the use of feathery wings to generate drag-based propulsion has been studied previously in thrips. In summary, any implication that the authors have identified a fundamentally new aerodynamic mechanism is simply not true, despite that fact that this point is implied in the title and elsewhere.

3) The authors second claim, that feathery wings evolved to minimize inertial cost in flight is a very interesting hypothesis that deserves serious consideration. This argument is entirely based on the estimates provided for the mass of the animal’s actual feather wing, as compared that the mass of the wing it would have if that wing was membranous. As stated above, this calculation is so essential, that it should not be delegated to the Supplemental Information. This critical section (on page 5 of the SI) reads,

“The mass and the moment of inertia of the equivalent membranous wing have been determined under the assumption that the membrane thickness is equal to the minimal thickness of the wing obtained by of (sic) measuring of cross sections of the wing obtained from histological preparations. Thus we prescribed the equivalent membrane thickness as a constant value of 0.98 microns.”

I have read and re-read the first sentence of this section many times and I cannot make sense of it. If the histological sections are those of a feather wing as mentioned in prior paragraphs, then what is the minimal thickness being measured? Are the authors taking the minimal section of the bristles? Are they referring to the minimal sections of the central blade of the wing? The given values for these structures (0.00388R and 0.008R respectively, assuming $R \sim 500$ microns) do not recover the given value of 0.98 microns for the virtual membranous wing. The given value for wing mass (0.421 micrograms)—here is a prime example of where the 3-digit precision of the measurement seems unjustified—may be a reasonable estimate, but insufficient information is provided to the reader to evaluate its accuracy. One thing the authors might do is compare their estimate for virtual wing thickness to that of a similarly-sized membranous-wing insect such as *Encarsia*, or even better, the thickness of the *Hydraena* sp. wing that members of the team have published in their recent *Arthropod Structure and Development* paper. Having seen that AS&D paper before, I was also expecting that the authors might provide a more detailed discussion or additional calculations that might explain the changes in wing morphology observed across the clade as a function of size. (For example, it is interesting that in that paper you observed wings that are intermediate between membranous and feathered.) In any event, so much depends upon these wing mass calculations that they ought to be described with greater transparency and moved to the main text of the paper.

Minor Issues

Line 35. The last sentence in the abstract about micro air vehicles is a non sequitur that has no place in the manuscript.

Line 61. The cited by Cheng and Sun is a preprint; the citation should be replaced with the actual publication in *Physics of Fluids*.

Line 99. Providing values for Reynolds number with a precision of 9.3 and 19.6 indicates a

misunderstanding of the basic concept of dimensionless numbers.

Line 122. Airflow visualization. The team did not perform flow visualization, that implies an experimental observation of the flow using techniques such as DPIV. The wording should be changed to indicate that you are discussing the flow patterns predicted by CFD simulations.

Line 154. The estimated wing mass.... Here is the point that the information in the SI should be revised for clarity and moved to the main text.

Line 155. 'outgrows'. Do you mean outgrowths?

Line 155. 'equivalent drag of 0.0421' How is this known to three significant digits?

Line 166. How, precisely, did you estimate mechanical power? There are several ways by which this might have been done.

Figure 3. No estimates of error are provided for the estimates of power in panel H.

Line 170. 'the mechanical power of the membranous wing would be dominated by inertia.' What is the basis for this claim? What are the estimated values of aerodynamic power and inertial power?

Line 172. Where does the value of 204 W/kg come from? Does this model assume that all inertial power (both positive to accelerate wing and negative to decelerate the wing) cost energy?, or can energy required to accelerate the wing be recovered as aerodynamic power in second half of stroke?

Line 215. '(II) The inertial forces are proportional to acceleration...' Isn't this point known a priori? How is this an observation from the study? The same could be said for item (III), which just falls from basic physics. Maybe the problem is that you should not use the term 'observations' on line 214, but rather say something like 'important principles related to our observations'.

Line 227. 'We arrive at the following....' Assuming you are correct about the reduction in wing mass allowed by ptiloptery, I honestly think there may be a better way of expressing your main (and very interesting finding) that does not rely so heavily on the power limits of muscles. After all, there are examples of insect muscle that can contract at a much faster frequency and deliver more power. My suggested alternative way of viewing things is that at low Reynolds numbers, the performance of solid, membranous wing barely outdoes that of feather wings in generating a drag-based vertical force. Thus, the tiny advantage in using a membranous wing is outweighed by the advantage gained in reducing inertial power by minimizing wing weight. This trade-off of large energy savings for small deficit in aerodynamic performance is only available at $Re \sim 10$ or lower, where the drop in leakiness is sufficient for feather wings to work. This is why one doesn't observe feather wings in larger species.

One last point: You never mention possible alternative hypotheses that might serve as alternative explanations or non-mutually exclusive hypotheses for ptiloptery. It strikes me that in addition to the power reduction arguments you are trying to make, there may also be a benefit to feathery wings in that they are more durable at a small size and with the ptiliid life style. Membranous wing might be much easier to tear and damage at this scale.

Referee #3 (Remarks to the Author):

This is an interesting and well-reported study of some fascinating tiny beetles. The manuscript brings a new understanding to some of the smallest flying animals. These insects have been very hard to study in this manner until now, mainly for imaging and computational reasons, but the authors have done an excellent job.

I agree with their claims that previous work on this topic has been limited in one way or another. While some of the main conclusions have been reported before (such as high angle of attack power strokes being a more drag-based method of weight support, the force profile being dominated by peaks during the power strokes, muscle power being a limiting factor, bristle wings reducing inertial costs), I believe this is genuinely novel work of the highest quality and represents a very significant step forward. This is the most comprehensive and realistic analysis of bristle wing flight to date.

There are a few issues that I would like to see fixed:

-Title. Do they 'excel', or do they merely 'manage'. You have made a strong case that they are severely limited in acceleration due to amplitude being maxed out and the muscle power increases that come with frequency changes.

-I recommend deleting the sentence about MAVs. The manuscript stands alone without reference to robotics and it will be a long time before any insight presented here can be incorporated into the design of any miniature vehicles.

-Unfortunate style issue: "length of about 200 μm^3 " The third reference makes length look like a unit of volume.

-It is not sufficiently precise to compare the body lengths of this featherwing species with "their relatives the Staphylinidae" for two reasons: The first is that there are tens of thousands of species of Staphylinidae, with varying sizes. The second is that the Staphylinidae are mostly notably for their elongated bodies, while the Ptiliinae family of Staphylinidae are not, so the comparison of speed or acceleration performance based on "a threefold difference in body length" is not particularly fair. It would be more robust to pick a species, or at least family, against which to compare.

-Is the frequency unexpectedly low for the size of insect? Does this give them some kinematic freedom to increase frequency when needed?

-How is the radius of gyration calculated? Where is the centre of rotation during the power strokes? It must be between the hinge and wing tip. Is it far outboard of the hinge? This location gives an idea of how close to rowing they are operating.

-Supplementary Fig 10: insert needs fixing on panel b.

-I would have appreciated more detail (justifying the claim that bristle wings are lighter) in lines 154-160 in this manuscript rather than referring to Reference 22. How leaky are the wings? What is the Re of the bristles? The petiole and setae must be quite stiff/reinforced if they deform so little during the flap – perhaps more so that the veins of a membrane wing which are stiffened by tension in the membrane.

-Line 168. I don't think an absence of elastic storage is 'remarkable'. This is a highly damped system which is, in some ways, ideal because allows the beetle to do a good job of converting all the wing KE into wake KE. Since the wings are light, what is the point of storing energy elastically? With that in mind, the continuously positive mechanical power requirement doesn't so much "compensate" for viscous losses but, rather, is enabled by the viscosity.

-I would like to see the final mesh around the setae and the local flows in those regions depicting a) the leakiness, b) separation around the wing perimeter during the power strokes, and c) the boundary layer during the feathered recovery strokes. These images would add a lot to the Supplementary information if they cannot be squeezed into the main body.

Author Rebuttals to Initial Comments:

We are grateful to you and to all Referees for finding our manuscript interesting and for suggesting how to improve it. We have now addressed all points raised by the Referees by revising our manuscript and the supplementary material. Our point-to-point responses to each of the three reviews are given below.

Referees' comments:

Referee #1 (Remarks to the Author):

This paper is a tour-de-force combination of 3D kinematics obtained from multi-camera high-speed-video of free-flying *Paratuposa* beetles together with computational fluid dynamics modelling to analyse the flight of tiny feather-winged beetles. The technical challenge in just obtaining the video is immense, the analysis is ground-breaking and the results are fascinating. I recommend publication. I have only a few minor suggestions relating to power calculations.

The authors show by calculation that the light weight of the feather wings compared to normal membranous wings reduces inertial costs of flapping. The calculations on power requirements are expressed as watts per kilo of bodyweight. I am deeply impressed that *Paratuposa* manages to fly at 28W/kg average (lines 161-1740. Professional cyclists manage about 6W/kg (the tour-de-france is imminent), the comparison might be worth making for general readers, and a *comparison with other flying animals would be useful to put these numbers in context*. However, *what I would really like to know is the power output of the beetle's muscles required for flight. Peak muscle power output is around 200W/kg (aerobic), that seems likely to be challenged by these beetles - which would further emphasise the author's correct claim that feathered wings are necessary for flight at such small sizes to minimise inertia. What is the average and instantaneous peak power output of the muscles, in W/kg?*

It must be possible to estimate muscle mass, or at least thoracic mass, from the SEM images.

We are grateful for such an impressive comparison and for the interesting question. We performed three-dimensional computer reconstruction of the *Paratuposa* flight muscles based on the CLSM and established the volume of the main muscles. The results and discussion are included, in a condensed form, in SI 18.

The muscle specific power per unit of body mass is an interesting characteristic for comparing different animals, but it is difficult to interpret, since it is influenced by many factors: the relative volume of the muscles, the frequency of their contractions, synchronous or asynchronous muscle type, muscle efficiency, elastic storage of inertia and metabolic input.

It is somewhat easier to operate with specific power per unit of muscle mass when comparing animals of the same group (i.e., insects). According to our data, the specific power of *P. placentis* flight muscles is 350 W/kg at a stroke frequency of 171 Hz, which coincides with the mass-specific power output of asynchronous muscles predicted by Pennycuik & Rezende and adopted by Ellington (<https://doi.org/10.1242/jeb.115.1.293>). *Bombus* sp. at a swing frequency of 150 Hz has a specific power of 56 W/kg (per body mass) and 186 W/kg (per muscle mass) (<https://doi.org/10.1242/jeb.148.1.53>), however, the latter value can be not entirely accurate due to approximate measurements of relative muscle mass. The high muscle power of *P. placentis* is possibly related to the overestimation of elastic storage in previously studied insects, as a result of which the power of their muscles was underestimated. Either the efficiency of the *P. placentis* flight musculature is increased due to the scale effect, since the muscle strength is proportional to the

section (linear dimensions squared) and the mass to volume (linear dimensions cubed). In order to verify these assumptions, additional studies of the functional morphology and ultrastructure of the musculature of the pterothorax of Ptiliidae are needed.

More could be made of the fact that at $Re < 10$ viscosity dominates, and these beetles are flying in the same regime as tiny swimmers like daphnia and diving beetles which also use feathered propulsory.

Indeed, it is interesting to compare the flight of *P. placentis* with the swimming of miniature aquatic crustaceans in the context of their similarity in Re values. We have added this comparison to the text of the manuscript.

Figure 2 A (the video sequence) is blurry and the supplementary video is much clearer, can anything be done to enlarge and clarify the images of the beetle wings? The clap is not readily discernible.

We have done our best to improve the images as much as possible: in the new version, frames with the original colour depth are used, scaling and smoothing have been performed, and levels are set more accurately. The video recordings themselves appear a little sharper, because in motion it is more difficult to notice imperfections in image detail.

These points are minor. I recommend publication.

Referee #2 (Remarks to the Author):

In their manuscript, Farisenkov and coworkers present the results of a morphological, kinematic, and aerodynamic analysis of a ptiliid beetles, which are among the tiniest of all flying insects. The Ptiliidae are known for the feather-like wings that the smallest members of the group possess. The research team has managed to collect high speed video of sufficient quality to perform a kinematic and aerodynamic analysis, thereby providing the first quantitative look at the flight biomechanics of this noteworthy group of small insects. While feather wings are not unique to the Ptiliidae, the results of this manuscript have the potential to answer interesting questions regarding the evolution of this morphological specialization that certainly finds an extreme within this group.

The paper makes two major claims about the flight biomechanics of *Paratuposa placentis*, the first is that it executes a so-called ‘novel’ flight style in which each half stroke execute a downward directed, drag-dominated power stroke followed by a rapid upward recovery in which the wing is rotated so as to produced less drag. The resulting full stroke produces two prominent upward force peaks per cycle. The second main claim is that due to the low mass of the wing, the inertial costs of flight (i.e. the cost to accelerate the wing mass back and forth) are extremely small, such that nearly all mechanical power cost goes into the production of aerodynamic power, and would so even in the absence of an elastic storage mechanism to reduce inertial power. In this view, ptiloptery evolved as a means of lowering flight costs by massively reducing wing mass—an option that is only available to insects operating at low Reynolds numbers where they can make use of the reduced leakiness of wings formed of arrays of fine hairs.

While I am impressed with the comprehensive nature of the manuscript, which relies on experts across an array of different disciplines and the fascinating look at the flight biology of these tiny creatures, I believe that the manuscript as it currently stands contains several critical flaws that precludes its publication in *Nature* at this time. I describe these issues below with the hope the authors can institute a substantive set of revisions that might render the paper more accurate and suitable for publication.

Major Concerns

1) *The quality of writing is generally poor compared to what is expected for a publication in Nature. The paper is replete with critical claims and calculations (most critically, the calculations of wing mass) that are buried in the supplemental material rather than being presented transparently in the main text. Having read both the main text and supplemental material very carefully, I feel that the authors have done a poor job deciding what material belongs in the main text and what belongs in the supplement. This lack of clarity starts with the very title of the paper, ‘A novel flight style allowing the smallest featherwing beetles to excel’. This title is vague and tautological—any animal that is not extinct could be said to ‘excel’—and does not summarize the main claim in the paper, which is that feather wings evolved in the Ptiliidae to minimize flight cost due to a reduction in inertial power requirements.*

We have corrected the title, taking into account the comments of all Referees, and considerably revised the text of the manuscript. We agree that the statement about the importance of reducing the mass and thus reducing the inertia of the wing is one of the main, and we paid more attention to it in the manuscript and the Supplementary Information.

Another issue that appears throughout the paper is a rather flagrant disregard for proper use of the concept of significant digits within nearly all calculations. Many numbers are provided with decimal value precisions that are difficult to justify given the precision of the measured values or constants upon which they are based.

We thank the Referee for this comment. We have carefully checked the numerical values. The accuracy (number of significant digits) of the values that are based on direct measurements, and therefore have a certain level of error, has been revised and, if necessary, reduced. Below we provide a justification for the accuracy of the main values as they are given in the new version of the manuscript:

The wing length was measured using micrographs, at a scale of 0.31 $\mu\text{m}/\text{pixel}$; the boundaries of the object are seen quite clearly during microscopy; the size of the object is about 1500 pixels. Thus, the accuracy of measuring the wing length up to 1 μm (3 digits) is valid.

The mass of the bristled wing was calculated on the basis of a chitin density of 1200 kg/m^3 , the volumes of the petiole and wing blade (obtained using confocal microscopy) and the volume of the setae, three-dimensional models of which were made from flat SEM images. Taking into account the accuracy of the density value of chitin (2 digits), we present the value of the mass of the bristled wing and of the hypothetical membranous model with an accuracy of 2 digits in the revised text of the manuscript.

The wingbeat frequency was calculated from high-speed video recordings; in each recording, we measured the number of frames during which several full strokes occur (on average, 8 strokes, about 200 frames). In total, the number of the frames is about 2500; this allows us to indicate the average frequency of strokes with an accuracy of 3 digits.

The air density was not measured by us, but was taken from a reference book, it is therefore given with an accuracy of 4 digits.

Please note that the “Input and output parameters of the CFD simulations specific to individuals” in the Supplementary Information indicates the frequency of strokes with an accuracy of 4 digits, as well as some other parameters. This does not reveal the true accuracy of measurements of the kinematic characteristics of the flight of the selected insects, but it is necessary for a clearer and more transparent description of the kinematic protocol for CFD.

2) The title of the paper and text throughout implies that *P. placentis* uses a novel form of flight kinematics and aerodynamics, in which vertical force is generated by a downward plunge (at high aerodynamic angle of attack) at the start of each upstroke, followed by an upward recovery stroke (with low aerodynamic angle of attack). However, textbooks on animal flight are literally replete with sections entitled ‘Drag-based propulsion’ or ‘Drag vs. Lift as a way to get thrust’, as well as illustrations of birds and insects employing a figure-of-eight pattern of wing motion. The kinematics of *P. placentis* are extreme to be sure, but there is

nothing unprecedented or unexpected in what they do compared to prior studies. Studies of hovering dragonflies have identified a drag-based mechanism in which the wing produces primarily drag during the downstroke and little reverse drag during the upstroke. Many authors have noted the U-shaped curve traced by the path of hovering insects (e.g. any one of the many published studies of Drosophila) and how the first portion of each downstroke and upstroke (more properly, forward stroke and backward stroke) contribute substantially to vertical force due to the downward motion and large aerodynamic angle of attack of the wing at that time. Most notably, Cheng and Sun published a paper this year (Physics of Fluids 33, 021905) on the kinematics and aerodynamics of Encarsia formosa, which execute a pronounced U-shaped pattern of wing motion that generates to prominent vertical force peaks at a Reynold number of ~10 (almost exactly the same Re as P. placentis). The describing those results, Cheng and Sun wrote, “Therefore, very small insects must use a “new” flapping mode.” I would probably qualify their statement to say that very small insects use a more distorted form of an “old” flapping mode, but the point is that neither the kinematics described here for P. placentis nor the drag-based mechanism that result can be accurately be described as ‘novel’. The authors themselves have reproduced the force traces reported in a prior preprint of the Cheng and Sun paper in Figure 4. Surely, the fundamental similarity between the results garnered from E. formosa and P. placentis and are more striking and noteworthy than the small differences, which are just as likely to have emerged from subtle inaccuracies in kinematic measurements or CFD simulations than from any fundamental difference in aerodynamics. Put plainly, the pattern of motion identified in P. placentis is not fundamentally different (or unexpected) from what has been recorded from a small insect using primarily solid wings. Further, as the authors note, the use of feathery wings to generate drag-based propulsion has been studies previously in thrips. In summary, any implication that the authors have identified a fundamentally new aerodynamic mechanism is simply not true, despite that fact that this point is implied in the title and elsewhere.

We agree with the Referee that we have not identified any “fundamentally new aerodynamic mechanism” akin to the Magnus effect or to Lighthill’s clap and fling. There is no such claim in our manuscript. But we do claim that, in *P. placentis*, we have identified a novel flight style: its wing motion is drastically different from any other known insect kinematics:

- This is not “normal hovering”, because deviation from the stroke plane is of the same order of magnitude as the flapping amplitude.
- This is not “rowing” or “drag-based propulsion” known from earlier textbooks. Here, lift and drag both contribute to the vertical force.
- This is not a U-shaped pattern. The figure of eight pattern of *P. placentis* stands out from the trend towards deeper U-shapes that has been proposed by Lyu et al. 2019 (<https://doi.org/10.1103/PhysRevE.99.012419>).

The unique wing-tip trajectory of *P. placentis* was a starting point for our analysis, which revealed that this insect uses an unusual combination of aerodynamic mechanisms:

- The wings clap (or near-clap) twice per wingbeat cycle: there is a dorsal and a ventral clap.
- Upstroke and downstroke are subperpendicular and aerodynamically much more symmetric than in any other studied small insect.
- Despite the low Reynolds number, lift and drag both contribute to the vertical force.

Our analysis shows that the extremely large excursion of the wing is advantageous only if the wings are very light, and bristled wings are light enough.

There is yet another complication: as long as the wings clap dorsally and ventrally, the flapping amplitude is maxed out, which means that the mean positional angle cannot be used for body pitch control! Instead, *P. placentis* uses the elytra for body pitch stabilization.

Thus, the novel flight style of *P. placentis* is a synergy of wing kinematics, bristled wing morphology, and elytron movements.

3) The authors second claim, that feathery wings evolved to minimize inertial cost in flight is a very interesting hypothesis that deserves serious consideration. This argument is entirely based on the estimates provided for the mass of the animal's actual feather wing, as compared that the mass of the wing it would have if that wing was membranous. *As stated above, this calculation is so essential, that it should not be delegated to the Supplemental Information.* This critical section (on page 5 of the SI) reads, "The mass and the moment of inertia of the equivalent membranous wing have been determined under the assumption that the membrane thickness is equal to the minimal thickness of the wing obtained by of (sic) measuring of cross sections of the wing obtained from histological preparations. Thus we prescribed the equivalent membrane thickness as a constant value of 0.98 microns."

*I have read and re-read the first sentence of this section many times and I cannot make sense of it. If the histological sections are those of a feather wing as mentioned in prior paragraphs, then what is the minimal thickness being measured? Are the authors taking the minimal section of the bristles? Are they referring to the minimal sections of the central blade of the wing? The given values for these structures (0.00388R and 0.008R respectively, assuming R~500 microns) do not recover the given value of 0.98 microns for the virtual membranous wing. The given value for wing mass (0.421 micrograms)—here is a prime example of where the 3-digit precision of the measurement seems unjustified—may be a reasonable estimate, but insufficient information is provided to the reader to evaluate its accuracy. One thing the authors might do is compare their estimate for virtual wing thickness to that of a similarly-sized membranous-wing insect such as *Encarsia*, or even better, the thickness of the *Hydraena* sp. wing that members of the team have published in their recent *Arthropod Structure and Development* paper. Having seen that AS&D paper before, I was also expecting that the authors might provide a more detailed discussion or additional calculations that might explain the changes in wing morphology observed across the clade as a function of size. (For example, it is interesting that in that paper you observed wings that are intermediate between membranous and feathered.) In any event, so much depends upon these wing mass calculations that they ought to be described with greater transparency and moved to the main text of the paper.*

Thanks for the comments and for the important questions. We have considerably revised this section in the manuscript, moved the most important parts into the main text and Methods, and also significantly improved the corresponding section in the Supplementary Information. In the initial calculations of the mass of the hypothetical membranous wing, we used the minimum thickness of the membranous part of the wing (not the setae) of *Paratuposa*, which was measured on cross-sections of the wing. In the new version, we additionally used measurements of the minimum wing thickness of several miniature insects: one of the smallest membranous-winged insects *Trichogramma telengai* (Hymenoptera: Trichogrammatidae, body length 0.45 mm), one of the smallest membranous-winged beetles *Orthoperus atomus* (Coleoptera: Corylophidae, 0.8 mm) and one of the smallest membranous-winged beetles of a family closely related to Ptiliidae — *Limnebius atomus* (Coleoptera: Hydraenidae, 1.1 mm). The Referee suggested using the thickness of *Encarsia* wings, but *Encarsia* is much larger than *Paratuposa*, so we considered it correct to take *Trichogramma*, one of the smallest representatives of Chalcidoidea, which has membranous wings with relatively short setae around the perimeter. The Referee also suggested using *Hydraena* sp., but this genus comprises relatively large representatives of the family Hydraenidae, and we measured instead one of the smallest representatives of this family, a species of the genus *Limnebius*. As before, to calculate the membranous counterpart of the *Paratuposa* wing, we used the minimum wing thicknesses of the above-mentioned insects, so that in our comparison the calculated membranous wing mass was estimated at the lower boundary. To verify our calculations, we also used the allometric model for scaling the wings and estimating the wing mass in insects of different sizes, which confirmed our comparisons of the calculated masses of the bristled wing of *Paratuposa* and its membranous counterpart (SI 3)

Minor Issues

Line 35. The last sentence in the abstract about micro air vehicles is a non sequitur that has no place in the manuscript.

We agree with this comment, and have deleted the mention of MAV from the manuscript.

Line 61. The cited by Cheng and Sun is a preprint; the citation should be replaced with the actual publication in *Physics of Fluids*.

The new paper by Cheng & Sun, *Physics of Fluids* 33, 021905 (1021) focuses on forward flight with the advance ratio $J = 1/(2 \cdot \phi \cdot R \cdot f) \approx 0.3$. The same authors discussed hovering ($J \approx 0.11$) of the same species in an earlier paper Cheng & Sun, *Journal of Fluid Mechanics* 855, 646–670 (2018). In our present study, we also focus on slow flight with $J \approx 0.1$. For this reason, we refer to the 2018 JFM paper, which is more relevant, rather than 2021 *PoF*.

The force coefficient in Fig. 4a of our original manuscript (now moved to the SI 18) corresponds to data from Lyu, Zhu & Sun, *Physical Review E* 99, 012419 (2019), Fig. 14d, individual EF4. This *E. formosa* individual produced two even peaks of the vertical force, which look similar to *P. placentis*. We digitized the plot from Lyu, Zhu & Sun and presented the data so as to show the dimensionless force coefficient (the original figure was the dimensional force) and so that $t = 0$ corresponds to the beginning of a downstroke (in the original figure, $t = 0$ was the beginning of an upstroke). However, in the revised supplementary information, we replaced this with the data for EF1 from Cheng & Sun, *Journal of Fluid Mechanics* 855, 646–670 (2018), because the latter also contains aerodynamic power data that we analyze in the revised Supplementary Information.

Line 99. Providing values for Reynolds number with a precision of 9.3 and 19.6 indicates a misunderstanding of the basic concept of dimensionless numbers.

We agree with the Referee that rounding Re to 9 and 20 in the main text would be appropriate. We have revised the text accordingly. However, we have to stress that the basic concept of dimensionless numbers does not disallow three-digit precision.

Specifically, the Reynolds number, besides a well-known physical interpretation as the ratio of scales of inertial over viscous stresses, has a precise mathematical definition. It is the inverse of the dimensionless factor ν/VL that appears in front of the viscous term of the incompressible momentum equation after nondimensionalizing the latter with a given length scale L and velocity scale V . In our case, L is to equal the radius of gyration (R_g) and $V = 2 \cdot \phi \cdot f \cdot R_g$. As long as the values of ϕ , f and R_g are prescribed with sufficient precision, Re can have three or more significant digits. In fact, in the flapping-flight research literature, it is common to show three or more significant digits in the Reynolds number (e.g., Table 1 in Usherwood & Ellington, *J. Exp. Biol.* 205:1565–1576, 2002 or Table 1 in Cheng & Sun, *J. Fluid. Mech.* 855:646–670, 2018). This may not be necessary, but this is not a mistake. Moreover, there are situations where Re must be shown with more than three significant digits. In studies on hydrodynamic instability it is justified to show six significant digits (e.g., Matharu, Hazel & Heil, *J. Fluid Mech.* 918:A42, 2021).

Line 122. Airflow visualization. The team did not perform flow visualization, that implies an experimental observation of the flow using techniques such DPIV. The wording should be changed to indicate that you are discussing the flow patterns predicted by CFD simulations.

We agree. To indicate that we are discussing the flow patterns predicted by CFD simulations, we have replaced “the air flow visualization” with “the air flow simulation visualized”.

Line 154. The estimated wing mass.... Here is the point that the information in the SI should be revised for clarity and moved to the main text.

We have followed this advice. In the revised manuscript, the “Wing mass and moments of inertia” section in the Methods section outlines the wing mass estimation procedure, the essential results of the wing mass and moment of inertia calculations are in the “Bristled vs membranous wings” section of the main text, and more details are given in the Supplementary Information.

Line 155. ‘outgrows’. Do you mean outgrowths?

Indeed, we meant outgrowths. We have corrected this.

Line 155. ‘equivalent drag of 0.0421’ How is this known to three significant digits?

The accuracy of our calculations is such that this value can be shown with two significant digits. We have revised the sentence accordingly.

Line 166. How, precisely, did you estimate mechanical power? There are several ways by which this might have been done.

We have included a new section 14 in the Supplementary Information, under the title “Mechanical power calculation”. We have also included a reference to it in the main text. The mechanical power has been calculated using the same algorithm as in Engels et al. (2016) *SIAM Journal on Scientific Computing*, but the Supplementary Information accompanying this article contains some additional clarification on the underlying mathematical model.

Figure 3. No estimates of error are provided for the estimates of power in panel H.

We have revised Fig. 3h so that now it shows a range of peak power estimates of the membranous wing. This quantifies the modelling uncertainty. The accuracy of the CFD simulation is addressed in the Supplementary Information, sections “CFD results” and “Numerical convergence”. We refer to them in the Methods section. While numerical convergence checks are mandatory in the CFD simulation, it is still not common to plot error bars. With as many input parameters as we have in our problem, and given the cost of one simulation, it is impractical to apply stochastic methods of uncertainty quantification.

Line 170. ‘the mechanical power of the membranous wing would be dominated by inertia.’ What is the basis for this claim? What are the estimated values of aerodynamic power and inertial power?

The Referee is right to question this claim. Indeed, for the membranous wing, we did not plot the inertial and aerodynamic power separately. The revised Fig. 3e shows the time evolution of these two quantities, and makes it clear that the inertial and the aerodynamic power are of the same order of magnitude, but the inertial power does not dominate. Therefore, we have revised the sentence in question as follows: “For a membranous wing, the inertial power compares in peak magnitude with the aerodynamic power.”

Line 172. Where does the value of 204 W/kg come from? Does this model assume that all inertial power (both positive to accelerate wing and negative to decelerate the wing) cost energy?, or can energy required to accelerate the wing be recovered as aerodynamic power in second half of stroke?

The value of 204 W/kg was the peak value of the body-mass-specific mechanical power, i.e., the maximum value during the wing beat cycle. The mechanical power is defined in the “Mechanical power calculation” of the Supplementary Information as the scalar product of the angular velocity of the wing and the actuation torque.

Simply put, the mechanical power is the torque times the angular velocity. This quantity varies in time, and it reaches 204 W/kg in the middle of a power stroke. No assumption about the nature of the actuation torque is made at this stage. The mean (i.e., averaged over one flapping cycle) values of the mass specific mechanical power in Fig. 3h assume so-called perfect elastic energy storage, which means that energy is harvested by the skeletomuscular system at times when the mechanical power is negative.

Some part of the energy required to accelerate the wing is transferred into the kinetic energy of the wing (as soon as the wing is accelerated). After that, the wing gets decelerated by the aerodynamic force — the kinetic energy of the wing is transferred to the energy of the fluid motion. Therefore, the energy required to accelerate the wing can be partly recovered as aerodynamic power in the second half of the stroke.

Line 215. ‘(II) The inertial forces are proportional to acceleration...’ Isn’t this point known a priori? How is this an observation from the study? The same could be said for item (III),

which just falls from basic physics. Maybe the problem is that you should not use the term ‘observations’ on line 214, but rather say something like ‘important principles related to our observations’.

We agree with the Referee that “observation” is not the right term in this context. We have followed the suggestion and replaced “observations” with “principles related to our observations”.

Line 227. ‘We arrive at the following....’ Assuming you are correct about the reduction in wing mass allowed by ptiloptery, I honestly think there may be a better way of expressing your main (and very interesting finding) that does not rely so heavily on the power limits of muscles. After all, there are examples of insect muscle that can contract at a much faster frequency and deliver more power. My suggested alternative way of viewing things is that at low Reynolds numbers, the performance of solid, membranous wing barely outdoes that of feather wings in generating a drag-based vertical force. Thus, the tiny advantage in using a membranous wing is outweighed by the advantage gained in reducing inertial power by minimizing wing weight. This trade-off of large energy savings for small deficit in aerodynamic performance is only available at $Re \sim 10$ or lower, where the drop in leakiness is sufficient for feather wings to work. This is why one doesn’t observe feather wings in larger species.

We thank the Referee for pinpointing the idea of the trade-off between aerodynamics and inertia, which was dilute in the original text. We re-phrased the sentences noted by the Referee and included them at the end of the “Bristled vs membranous wings” section:

“At low Reynolds numbers, impermeable membranous wings barely outperform leaky bristled wings in generating aerodynamic force. Thus, the small advantage in using a membranous wing is outweighed by the advantage gained in reducing inertial torques and power through minimizing wing mass. This trade-off of energy savings for a small penalty in aerodynamic force generation is only available at $Re \approx 10$ or lower, where sufficiently low leakiness can be achieved with a small number of slender bristles.”

However, we cannot go as far as claiming that “This is why one doesn’t observe feather wings in larger species”, even if this seems correct. This cannot be concluded without considering structural stiffness. There exist larger species with bristled wings and $Re \approx 100$ (e.g., adults of the Australian genus *Idolothrips* can have a body length greater than 10 mm and still have bristled wings, like all extant winged thrips). They have relatively shorter and thicker setae which are more densely spaced. In addition, the wing beat frequency scaling should be taken into consideration: it has an interesting plateau between $Re \approx 10$ and $Re \approx 100$. We are collecting data from other species that would allow to elaborate on these points.

One last point: You never mention possible alternative hypotheses that might serve as alternative explanations or non-mutually exclusive hypotheses for ptiloptery. It strikes me that in addition to the power reduction arguments you are trying to make, there may also be a benefit to feathery wings in that they are more durable at a small size and with the ptiliid life style. Membranous wing might be much easier to tear and damage at this scale.

Indeed, there have been alternative hypotheses proposed in the insect biomechanics community.

It is not impossible that bristled wings offer more advantages than just lower inertia. First, there is evidence that microtrichia on the wing membranes of large insects are needed for unwettability and self-cleaning with water droplets (Wagner et al., 1996 <https://doi.org/10.1111/j.1463-6395.1996.tb01265.x>). Wetting is an undesirable event as it can cause spontaneous folding (Dickerson et al., *Phys. Fluids* 27:021901, 2015 <https://doi.org/10.1063/1.4908261>). One can think of self-cleaning mechanisms of bristled wings along these lines. Second, static charge may build up at the tips of the bristles and it may prevent

dust and sand accumulation. Third, an important function of the veins of membranous wings is “crack termination”. Cracks can initiate in the membrane, but their early propagation stops behind a vein (Rudolf et al., *J. Mech. Behav. Biomed.*:99:127–133, 2019 <https://doi.org/10.1016/j.jmbbm.2019.07.009>). Bristled wings do not have this problem of crack propagation, but it is not self-evident if they are overall more damage-resistant.

However, to the best of our knowledge, the above alternative hypotheses for bristled wings remain speculative, as factual data and evaluation criteria are not available. We prefer not to include this discussion in the paper.

Referee #3 (Remarks to the Author):

This is an interesting and well-reported study of some fascinating tiny beetles. The manuscript brings a new understanding to some of the smallest flying animals. These insects have been very hard to study in this manner until now, mainly for imaging and computational reasons, but the authors have done an excellent job.

I agree with their claims that previous work on this topic has been limited in one way or another. While some of the main conclusions have been reported before (such as high angle of attack power strokes being a more drag-based method of weight support, the force profile being dominated by peaks during the power strokes, muscle power being a limiting factor, bristle wings reducing inertial costs), I believe this is genuinely novel work of the highest quality and represents a very significant step forward. This is the most comprehensive and realistic analysis of bristle wing flight to date.

There are a few issues that I would like to see fixed:

-Title. Do they ‘excel’, or do they merely ‘manage’. You have made a strong case that they are severely limited in acceleration due to amplitude being maxed out and the muscle power increases that come with frequency changes.

As noted above, we have corrected the title of the manuscript.

-I recommend deleting the sentence about MAVs. The manuscript stands alone without reference to robotics and it will be a long time before any insight presented here can be incorporated into the design of any miniature vehicles.

The mention of MAVs was removed from the manuscript in accordance with the wishes of the Referees.

-Unfortunate style issue: “length of about 200 μm^3 ” The third reference makes length look like a unit of volume.

It has been corrected.

-It is not sufficiently precise to compare the body lengths of this featherwing species with “their relatives the Staphylinidae” for two reasons: The first is that there are tens of thousands of species of Staphylinidae, with varying sizes. The second is that the Staphylinidae are mostly notably for their elongated bodies, while the Ptiliinae family of Staphylinidae are not, so the comparison of speed or acceleration performance based on “a threefold difference in body length” is not particularly fair. It would be more robust to pick a species, or at least family, against which to compare.

We are grateful to the Referee for the comment and have corrected the text to make the comparison more accurate. But the question of the body shape of beetles is not so unambiguous. On the one hand, many, especially small Staphylinidae, are not so elongated, on the other hand, many Ptiliidae, including *Paratuposa*, are rather strongly elongated. See below two pairwise comparisons of Ptiliidae and Staphylinidae having approximately the same average flight speed in each pair (all pictures are on the same scale). Even assuming the difference in the shape of the body, the beetles in these pairs differ several times in any measurement of size. The names and characteristics of the beetles are given in one of the publications we refer to (<https://doi.org/10.1073/pnas.2012404117>), and it seems to us inappropriate to provide a more detailed comparison in the Introduction to our article.

-Is the frequency unexpectedly low for the size of insect? Does this give them some kinematic freedom to increase frequency when needed?

Wingbeat frequency undoubtedly correlates with body size in insects. But it is associated with many factors, including wing loading. The ptiliids studied by us have relatively large wings; therefore, their wingbeat frequency is low, for instance in comparison with mosquitoes, which beat their wings at a frequency of several hundred Hz. However, for an accurate study of this relationship, there is still not enough data on the body masses of the smallest insects.

The range of wingbeat frequency in *P. placensis* could actually be wider than that obtained in our study. This possibility is indirectly indicated by the fact that the flight speeds of the studied individuals were significantly lower than the maximum flight speeds for this species that we obtained in our previous study (<https://doi.org/10.1073/pnas.2012404117>). These speeds were recorded in a large flight chamber, which allowed the beetles to accelerate, but the magnification and frequency of the video recordings did not allow measuring the wingbeat frequency. Nevertheless, it should be expected that at high flight speeds, beetles need to increase the frequency of their flapping, since they practically cannot increase the amplitude.

-How is the radius of gyration calculated? Where is the centre of rotation during the power strokes? It must be between the hinge and wing tip. Is it far outboard of the hinge? This location gives an idea of how close to rowing they are operating.

We have included a new section in the Supplementary Information to explain our calculation of the radius of gyration, and added a reference to it in the main text. Note that the quantity that we refer to is a geometrical property that is customarily used in the calculation of dimensionless numbers of flapping wings such as the Reynolds number and the aerodynamic force coefficients. It is not directly related to the instant centre of rotation.

The instant centre of rotation in our case is near the hinge. From this point of view, the *Paratuposa* flight style is not similar to rowing. A more extended discussion has been added in the Supplementary Information as a new section.

-Supplementary Fig 10: insert needs fixing on panel b.

We have corrected this.

-I would have appreciated more detail (justifying the claim that bristle wings are lighter) in lines 154-160 in this manuscript rather than referring to Reference 22. How leaky are the wings? What is the Re of the bristles? The petiole and setae must be quite stiff/reinforced if they deform so little during the flap – perhaps more so that the veins of a membrane wing which are stiffened by tension in the membrane.

We have re-examined the wing mass estimates, modified the “Bristled vs membranous wings” section in the main text to include more detail and added a new section (“Wing mass and moments of inertia”) in the Methods. The bristled wing mass is calculated as a product of the SEM image-based geometrical models and the assumed uniform material density. The measurement error of linear dimensions is of the order of 1%; therefore, the error of the internal volume calculation can be estimated as 3%. The standard deviation of wing cuticle density is of the order of 100 kg/m³ (Vincent & Wegst, *Arthropod Structure and Development* 33(3): 187–199, 2004). This suggests that the overall accuracy of the wing mass calculation is of the order of magnitude of 13%. For the mass of the equivalent membranous wing, we now provide a possible range, which is based on the range of membrane thickness measured in some of the tiniest membranous-wing insects. To cross-validate the calculations, we estimated the membranous wing mass by extrapolation of an allometric trend obtained from direct measurement, see Supplementary Information, section “Allometric analysis of wing mass”.

The aerodynamic properties of the bristled wing of *P. placentis* were studied in our previous work (<https://doi.org/10.1007/s00348-020-03027-0>). The overall leakiness at Re 9.9 is 0.24. Re of a bristle (seta) is in the order of 10⁻² to 10⁻¹, depending on the velocity.

The stiffness of a bristled wing is an interesting topic to study, given how small the wing deformations are in *P. placentis*. On the one hand, a decrease in the area of veins and of the membrane should weaken the wing; on the other hand, the linear dimensions of skeletal structures decrease allometrically during miniaturization in animals, since the load on them is proportional to the body mass (linear dimensions cubed), and the strength is proportional to the cross-sectional area (linear dimensions squared). Our data are still insufficient to compare the stiffness and deformations of membranous and bristled wings of microinsects; we need both data on the aerodynamic loads and measurements of deformations in flight, as well as a study of the structure of the wings: their geometry, the thickness of various sections, and the distribution of chitin and resilin in the wings. To date, only the structure of the wings of another, larger species of Ptiliidae, *Acrotrichis sericans*, has been studied (<https://doi.org/10.1038/s41598-020-73481-7>), but the accuracy of measuring the wing thickness by confocal microscopy turned out to be insufficient for modelling its mechanical properties.

-Line 168. I don't think an absence of elastic storage is 'remarkable'. This is a highly damped system which is, in some ways, ideal because allows the beetle to do a good job of converting all the wing KE into wake KE. Since the wings are light, what is the point of storing energy elastically?

With that in mind, the continuously positive mechanical power requirement doesn't so much “compensate” for viscous losses but, rather, is enabled by the viscosity.

We agree with the interpretation proposed by the Referee. In the main text, section “Bristled vs. membranous wing”, we revised the sentence beginning with “Remarkably, the total mechanical power...”, as follows:

“The total mechanical power of the bristled wing model (Fig. 3e) remains positive during the entire wingbeat cycle, because low inertia of the wing and high viscous damping of the surrounding air enable continuous energy transfer from the flight apparatus to the wake. No elastic energy storage is required.”

We have added to the supplement a new section, “Details of the flow around the setae”.

And we have revised the last sentence of the abstract accordingly.

Original manuscript:

“This novel flight style evolved during miniaturization may compensate for costs associated with air viscosity and helps explain how extremely small insects preserved superb aerial performance during miniaturization.”

Revised manuscript:

“These adaptations help explain how extremely small insects have preserved superb aerial performance during miniaturization, which was one of the factors of their evolutionary success.”

-I would like to see the final mesh around the setae and the local flows in those regions depicting a) the leakiness, b) separation around the wing perimeter during the power strokes, and c) the boundary layer during the feathered recovery strokes. These images would add a lot to the Supplementary information if they cannot be squeezed into the main body.

We have included an additional figure in SI 11 (“Visualization of the block-based adaptive grid used in the simulations”) and an additional Supplementary Information section (“Details of the flow around the setae”), with illustrations of the velocity, pressure and vorticity fields during the power and recovery strokes.

Reviewer Reports on the First Revision:

Referee #1 (Remarks to the Author):

The authors have done a good job of responding to the referees comments. In particular they have clarified the images and videos significantly, tightened up the text, and dealt well with the specific (and rather useful) points made by referee 3 and 4. The videos are now clear and dramatic. The paper is well written and clear. I recommend publication.

Referee #2 - did not submit a report

Referee #3 (Remarks to the Author):

[no further comments]

Author Rebuttals to First Revision:

We are very grateful for such a high assessment of our study. In the new improved version of our manuscript we have made all changes that you required. All technical documentation has been filled in and corrected.

Referees' comments:

Referee #1 (Remarks to the Author):

The authors have done a good job of responding to the referees comments. In particular they have clarified the images and videos significantly, tightened up the text, and dealt well with the specific (and rather useful) points made by referee 3 and 4. The videos are now clear and dramatic. The paper is well written and clear. I recommend publication.

Referee #2 - did not submit a report

Referee #3 (Remarks to the Author):

[no further comments]